# Prompt Engineering at Scale: Provably Effective Multi-Agent Cascades for Attribute Generation in E-Commerce

## Abstract

Developing specialized Large Language Model (LLM) prompts for domain-specific tasks at scale remains a significant hurdle, particularly for e-commerce applications managing tens of thousands of distinct product attributes. We introduce **CascadeAgent**, a novel multi-agent framework that automates prompt adaptation and specialization through semantic gradient-based refinement. CascadeAgent employs a hierarchical architecture where a central Prompting Agent orchestrates four specialized counterparts—Writing, Generation, Evaluation, and Flaw Detection—that collaboratively analyze domain metadata, construct attribute-specific prompts, and enhance performance through iterative feedback. Our approach combines Multi-pass Prompt Generation (MPG) for modularity with textual gradient optimization that refines instructions based on detected error patterns. We provide formal theoretical analysis demonstrating provable convergence towards reduced loss under defined conditions. In a large-scale e-commerce case study on product attribute enrichment, CascadeAgent generated and optimized over 27,000 distinct prompts, achieving improvements of +21% to +33% in precision and +12% to +14% in coverage across multiple LLMs.

These results highlight CascadeAgent's capacity for robust, automated prompt engineering at industrial scale, while making more affordable models viable for deployment. The framework's modular design, iterative improvement mechanism, and theoretical guarantees make it a strong candidate for applications requiring principled refinement of vast numbers of task-specific prompts.

## 1 Introduction

Enhancing product listings with detailed and accurate attribute information is a critical, yet challenging task in e-commerce. Automatically enriching product catalogs simplifies the listing process for sellers and significantly improves the shopping experience for customers by boosting search relevance, product discovery, and informed purchasing decisions. While Large Language Models (LLMs) offer a promising solution for automating this process, adapting them to domain-specific tasks remains challenging, as generic prompts often fail to capture the nuanced requirements of specific product attributes, impacting catalog quality and user experience.

The core complexity lies in the sheer heterogeneity of e-commerce catalogs: countless product categories, each with a unique set of relevant attributes, demand specialized handling. Effective attribute extraction requires instructions meticulously tailored to specific product-attribute (PA) combinations while ensuring consistency and quality across the entire catalog. To address this scale and specificity, we first introduce **Multi-pass Prompt Generation (MPG)**, a strategy that modularizes the problem by processing individual attributes with dedicated prompts. This approach allows for precise instruction tuning without creating overly complex prompts and provides a robust foundation for systematic, scalable optimization by isolating the refinement of each attribute.

While prior work has explored individual aspects of prompt engineering and multi-agent systems Shin et al. (2020); Yang et al. (2024); Ye et al. (2023); Yuksekgonul et al. (2024); Chang et al. (2024); Shinn et al. (2023), our key innovation lies in their principled integration for industrial-scale attribute extraction. Unlike previous approaches that handle dozens to hundreds of attributes Zheng et al. (2018); Yan et al. (2021c); Yang et al. (2022); Fang et al. (2024); Zhang et al. (2024); Gong et al. (2025), CascadeAgent's novel architecture enables management of over 27,000 attribute-specific prompts while providing theoretical guarantees of convergence. The significant performance improvements (+33% in precision, +14% in coverage) and ability to make affordable models competitive with premium ones demonstrate a fundamental rethinking of how to approach large-scale prompt engineering.

Building on MPG, we propose **CascadeAgent**, a novel multi-agent framework designed to automate the creation and, crucially, the iterative refinement of these attribute-specific instructions. CascadeAgent orchestrates five specialized agents—Prompting, Writing, Generation, Evaluation, and Flaw Detection—in a collaborative loop. This system begins by generating initial instructions based on catalog guidelines and seller data. It then enters iterative refinement cycles where outputs are evaluated, flaws are identified, and instructions are systematically updated using a semantic gradient-based optimization Shin et al. (2020). This process, which continues until desired accuracy is achieved or iteration limits are met, ensures that PA-specific instructions are progressively enhanced.

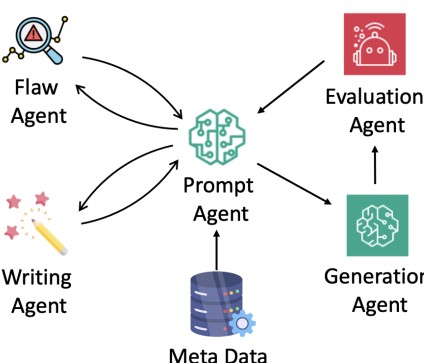

Figure 1: CascadeAgent with five specialized agents to optimize catalog enrichment.

A key contribution of this work is not only the empirical demonstration of CascadeAgent's effectiveness but also a **formal theoretical analysis** of its iterative refinement mechanism. We model the system's dynamics and prove, under reasonable assumptions, that the expected catalog loss decreases, providing principled validation for our design.

The contributions of our work are thus multi-faceted:

1. **Multi-pass Prompt Generation (MPG):** A scalable framework that decomposes complex catalog enrichment into attribute-specific sub-tasks, each managed by a specialized prompt.
2. **CascadeAgent:** A novel multi-agent system that collaboratively creates, evaluates, and refines attribute-specific instructions, incorporating domain knowledge (e.g., catalog guidelines, seller preferences) and optimized through an iterative, semantic gradient-based approach.
3. **Theoretical Guarantees:** A formal analysis demonstrating the convergence properties of CascadeAgent's iterative refinement process toward reduced catalog loss.
4. **Industrial-Scale Empirical Validation:** Demonstration of CascadeAgent's effectiveness in a real-world e-commerce setting, achieving up to +33% precision and +14% coverage improvements on attribute value generation across multiple LLMs.

## 2 METHOD

To address the challenge of scaling attribute extraction, we propose **CascadeAgent**, a novel framework featuring a multi-agent architecture that systematically generates, refines, and optimizes attribute-specific instructions. CascadeAgent is built upon two core concepts: MPG for modularity and an agentic workflow for initial prompt creation and iterative refinement via textual gradients.

## 2.1 MPG: A Scalable Framework for Attribute-Specific Processing

We decompose the complex catalog enrichment problem into a series of manageable, attribute-specific sub-tasks. This decomposition strategy—henceforth termed **Multi-pass Prompt Generation (MPG)** strategy—offers the following significant advantages:

1. **Modular Decomposition:** MPG decomposes the task into attribute-specific modules. Each attribute gets its own specialized prompt, allowing for precise instruction tuning without the complexity overhead of conditional logic.
2. **Independent Optimization:** By isolating each attribute's handling, MPG enables independent optimization of prompts. If a specific attribute's extraction is underperforming, its prompt can be refined without affecting the performance of others.
3. **Scalable Architecture:** The modular nature of MPG makes it inherently scalable. While previous approaches were either limited to handling small set of attributes or required a lot training data Tavanaei et al. (2024); Yan et al. (2021b), our approach successfully manages over 27,000 distinct product-attribute combinations.

Under the MPG framework, each product attribute is processed individually using a dedicated, specialized prompt. This process initiates with a generic base prompt that incorporates essential product information (e.g., title, description). It is then tailored to the specific attribute by integrating relevant attribute metadata, such as its expected data type or permissible values. This attribute-level approach enables independent optimization through the process described in the following section.

## 2.2 Collaborative Refinement: The CascadeAgent

Building upon MPG, we introduce **CascadeAgent**, a novel multi-agent system that automates the creation and refinement of attribute-specific prompts. CascadeAgent operates through a synergistic, iterative refinement loop, illustrated in Figure 1. The core philosophy is that prompts are not static artifacts but are dynamically "sculpted" by specialized agents, each contributing its expertise, much like a collaborative human team but operating at machine scale and speed to ensure optimal accuracy and relevance across diverse product categories.

Unlike previous approaches that rely on single-agent architectures or static prompt engineering techniques, the Agentic Cascade leverages a team of specialized agents working in concert to continuously improve prompt quality and performance. The workflow orchestrates five distinct agents in a continuous cycle, where each agent's output informs the next:

**Prompting Agent:** Serves as the central orchestrator. It initiates the cycle by obtaining relevant metadata (e.g., product category, attribute specifications) and generating an initial base prompt. It coordinates the activities of the other agents and integrates their outputs to drive the refinement process.

**Writing Agent:** Receives the current prompt and associated metadata. It synthesizes diverse information sources—ranging from broad catalog guidelines and attribute specifications (e.g., data types, valid values from category definitions) to nuanced seller preferences gleaned from historical data patterns—into coherent, context-aware instructions. This step is vital for grounding the prompts in the specific operational realities and quality standards of the e-commerce platform.

**Generation Agent:** Takes the refined instructions from the Writing Agent and executes the attribute value generation task. It processes input data, such as product titles and descriptions, using an underlying LLM to generate attribute values that are intended to be accurate and aligned with catalog expectations.

**Evaluation Agent:** Assesses the attribute values produced by the Generation Agent. It applies a set of pre-defined evaluation criteria, comparing generated values against ground-truth data or using other quality heuristics, to provide detailed feedback on the performance of the current instruction.

**Flaw Agent:** Performs a crucial diagnostic role. It analyzes the feedback from the Evaluation Agent to move beyond simple error counts, identifying systematic problems, common error patterns, or ambiguities in the generated outputs that indicate deficiencies in the current instruction. Its summarized findings are not just logs but actionable critiques that guide subsequent prompt rewriting.

The cycle then returns to the Prompting Agent, which, in concert with the Writing Agent, leverages the Flaw Agent's diagnosis to iteratively improve the instruction. This refinement is guided by a textual gradient-based optimization strategy, detailed in the next section. The process for each product-attribute (PA) prompt continues independently until it achieves a desired level of accuracy or a predetermined iteration limit is reached, allowing for a varying number of refinement cycles tailored to the complexity of each specific PA.

This multi-agent approach offers several advantages:

- **Specialization:** Each agent focuses on a specific aspect of the prompt engineering process, allowing for more nuanced and effective improvements.
- **Continuous Improvement:** The iterative nature of the cascade enables ongoing refinement, adapting to new patterns and edge cases over time.
- **Interpretability:** By breaking down the refinement process into distinct steps, the CascadeAgent provides insights into how and why prompts are being modified.
- **Scalability:** The modular architecture allows for parallel processing of multiple attribute-specific prompts, enabling efficient optimization at scale.

## 2.3 Principled Improvement: Optimization with Textual Gradients

To fully realize the potential of this multi-agent architecture at scale, we need a principled approach to systematic improvement across thousands of prompts simultaneously. We introduce an iterative refinement process detailed in Algorithm 1. Our approach builds upon semantic 'gradient' methodology from ProTeGi Pryzant et al. (2023), but adapts it specifically for large-scale attribute extraction. Unlike traditional optimization approaches that greedily pursue a single optimal solution, our method maintains a diverse pool of candidate instructions. This design choice provides two key benefits: resistance to premature convergence to local optima, and the ability to explore multiple promising refinement paths simultaneously—essential features when optimizing prompts across diverse attribute types. The process leverages rich textual feedback signals to guide refinement, enabling CascadeAgent to achieve high performance across diverse attribute types at industrial scale. Section 3 provides a formal analysis of the convergence properties of this optimization strategy.

---

**Algorithm 1** Optimization with Textual Gradients

**Input:**
    Initial LLM instruction $I_0$, Training data with errors $D$
    Number of minibatches $m$, sample size $n$
    Top-K selection size $K$, max iterations $T$
**Output:** Optimized LLM instruction
1: Initialize pool: $P \leftarrow \{I_0\}$
2: Initialize top-K: $best\_K \leftarrow \{I_0\}$
3: Initialize counter $t \leftarrow 1$
4: **repeat**
5:    $new\_instructions \leftarrow \emptyset$
6:    **for** each instruction $I$ in $best\_K$ **do**
7:      **for** each minibatch $B_i$ of training data **do**
8:       Sample $n$ errors: $S_i$
9:       $flaw\_summary \leftarrow \texttt{FlawAgent}(S_i)$
10:      $new\_I \leftarrow \texttt{RewritingAgent}(I, flaw\_summary)$
11:      Add $new\_I$ to $new\_instructions$
12:      **end for**
13:    **end for**
14:    Evaluate $new\_instructions$
15:    Update pool: $P \leftarrow P \cup new\_instructions$
16:    Select top-K: $best\_K \leftarrow$ Top-K from $P$
17:    $t \leftarrow t + 1$
18: **until** convergence or $t > T$
19: **Return:** Best instruction from $best\_K$

---

# 3 THEORETICAL ANALYSIS OF CASCADEAGENT

We present a formal theoretical analysis of CascadeAgent's convergence properties and performance guarantees.

**Modeling Prompt Refinement.** We model the iterative refinement of a single attribute-specific instruction ($\pi_t$ at iteration $t$) as a countable-state deterministic Markov Decision Process (MDP).

- *State* $s_t = \pi_t \in \Pi$: The current instruction.
- *Action* $a = g(\cdot, \varepsilon)$: A rewrite by the Writing Agent, guided by a summary $\varepsilon$ from the Flaw Agent.
- *Transition* $s_{t+1} = a(s_t)$: The deterministic outcome of applying the rewrite.
- *Reward* $r(s) = -\mathcal{L}(s) \in [-\kappa_{\max}, 0]$: The negative of the catalog loss for the current instruction $\pi$. The catalog loss $\mathcal{L}(\pi)$ is defined as the expected weighted sum of distinct error types:

$$\mathcal{L}(\pi) := w_{\text{val}} \Pr[E_{\text{val}}(\pi)] + w_{\text{omis}} \Pr[E_{\text{omis}}(\pi)] + w_{\text{commis}} \Pr[E_{\text{commis}}(\pi)].$$

Here, $\Pr[E_{\text{val}}(\pi)]$ is the probability of an *incorrect non-blank* prediction (value error) when using prompt $\pi$. Similarly, $\Pr[E_{\text{omis}}(\pi)]$ is the probability of an *omission* error (predicting blank when a value is expected), and $\Pr[E_{\text{commis}}(\pi)]$ is the probability of a *commission* error (predicting a value when blank is expected). The positive weights $w_{\text{val}}, w_{\text{omis}}, w_{\text{commis}}$ assign penalties to each error type, and are chosen to sum to 1. The maximum possible penalty for any single error instance is $\kappa_{\max} = \max(w_{\text{val}}, w_{\text{omis}}, w_{\text{commis}})$, and the minimum $\kappa_{\min} = \min(w_{\text{val}}, w_{\text{omis}}, w_{\text{commis}})$.

The objective is to find a policy $\varpi$ (a sequence of rewrite choices) that maximizes the discounted cumulative reward $V^{\varpi}(s) = \mathbb{E}_{\varpi}[\sum_{t=0}^{\infty} \gamma^t r(s_t)]$. CascadeAgent's Algorithm 1 implements a *textual-gradient greedy policy* ($\varpi_{\text{TG}}$), which selects rewrites that minimize empirical loss on minibatches and maintains a pool of Top-$K$ candidate instructions.

Our analysis establishes two crucial properties of this refinement process:

**(A) Reliable Candidate Selection (Top-$K$ Safety):** The Top-$K$ selection mechanism (Algorithm 1, line 16) is vital for exploring the instruction space effectively. Proposition B.1 shows that, with sufficient samples, this step retains the true best instruction from the candidate pool and ensures the best true loss does not increase, with high probability.

**(B) Progressive Error Reduction:** If rewriting instructions is, on average, more likely to fix errors than to introduce new ones, the catalog loss is expected to decrease. Proposition B.2 quantifies this.

Combining these insights, Theorem 1 demonstrates that CascadeAgent's policy $\varpi_{\text{TG}}$ indeed drives down catalog loss and improves the value function, up to limitations imposed by irreducible errors and estimation noise.

**Theorem 1** (Loss and Value Improvement of $\varpi_{\text{TG}}$). *Let $\vartheta_{\text{est}}$ be the failure probability of the Top-$K$ estimator (from Prop.B.1). Assume the marginal-churn condition $p_{\text{fix}} > p_{\text{break}} \geq 0$, where $p_{\text{fix}}, p_{\text{break}}$ pertain to the correction/introduction of any true error type (as defined for Prop. B.2). Define $\lambda = 1 - (p_{\text{fix}} + p_{\text{break}})$, $r = \kappa_{\max}/\kappa_{\min}$ (derived from penalties for true errors, where $\kappa_{\max}$ is the max penalty), and let $\phi = r\lambda$. Assume $\phi \in [0, 1)$.*

(i) ***One-step expected loss reduction:*** *The expected loss of the prompt at the next iteration, $\mathcal{L}_{t+1}$, given the current prompt's loss $\mathcal{L}_t$, satisfies: $\mathbb{E}[\mathcal{L}_{t+1} \mid s_t] \leq \phi \mathcal{L}_t + \kappa_{\max} p_{break} + \vartheta_{\text{est}}$.*

(ii) ***Horizon-bounded loss:*** *The expected loss at iteration $T$, starting from an initial loss $\mathcal{L}_0$, satisfies: $\mathbb{E}[\mathcal{L}_T] \leq \phi^T \mathcal{L}_0 + \dfrac{\kappa_{\max} p_{break} + \vartheta_{\text{est}}}{1 - \phi} (1 - \phi^T)$.*

(iii) ***Value function increase:*** *With a discount factor $\gamma \in (0, 1)$, the policy $\varpi_{\text{TG}}$ improves the value function $V^{\varpi_{\text{TG}}}(s_t)$ at each step where the current loss $\mathcal{L}_t$ exceeds the asymptotic floor $\mathcal{L}_\infty^\star = (\kappa_{\max} p_{break} + \vartheta_{\text{est}})/(1 - \phi)$. Specifically, $\mathbb{E}[V^{\varpi_{\text{TG}}}(s_{t+1}) - V^{\varpi_{\text{TG}}}(s_t) \mid s_t] \geq \dfrac{1 - \phi}{1 - \gamma\phi} [\mathcal{L}_t - \mathcal{L}_\infty^\star]^+$.*

*(Formal statement and proof are in Appendix B.3.)*

The theorem shows that CascadeAgent's loss contracts geometrically up to an irreducible error floor dictated by $p_{\text{break}}$ (inherent task difficulty) and $\vartheta_{\text{est}}$ (evaluation error). As long as rewrites are *net beneficial* and the estimator reasonably accurate the expected loss shrinks. The better the evaluation or the more amenable the domain to correction (smaller $p_{\text{break}}$), the lower this achievable loss floor.

## 4 Experimental Design and Evaluation

### 4.1 Dataset and Sampling

Our evaluation assessed the effectiveness of generating product attribute values using prompts crafted by our cascade measured against manually verified data. The process comprised two phases:

**Cascade initialization & evaluation set:** We setup the cascade to create **27,000** Product-Attribute (PA) specific LLM prompts (instructions) using catalog metadata. To evaluate at this massive scale, we designed both aggregate catalog level and fine-grained PA level evaluation sets. For catalog-level evaluation, we used a search weighted sample of **10,000** products with **304,847** attribute labels across **1394** product categories, reflecting real-world product distribution. For granular evaluation at the PA-level, we created an extensive dataset of **2,897 PA pairs** based on highest customer relevance, each with at least 100 test labels resulting in **304,000** labels in the set.

**Iterative optimization:** The iterative optimization of the cascade with textual gradients required at least 150 verified human labels per PA. We selected 1,879 PAs of highest customer relevance that met the label criteria, creating 1,879 PA high fidelity set. For each PA, the data set was split into training (50 samples) and validation (100 samples) sets corresponding to instruction refinement and top-K selection respectively. Additional holdout test (100+) samples) sets were used for performance evaluation.

### 4.2 Model Selection and Computing Infrastructure

We used two LLMs in our cascade: Claude 3.5 Sonnet for flaw detection and writing agents due to its superior capabilities in error analysis, and Mistral NeMo for attribute generation and evaluation due to its favorable inference cost profile. All experiments were conducted on a network with 6 AWS EC2 P5 instances with a total of 48 NVIDIA H100 GPUs (80GB each), achieving a throughput of 613 PTAs per hour through parallel processing.

### 4.3 Hyperparameter Configuration

Based on ablation studies (Appendix A.1), we optimized the system with 5 minibatches, 5 error cases per minibatch, and top K=3 selection, balancing optimal error diversity and generalization performance while maintaining computational efficiency.

### 4.4 Evaluation Methodology

We evaluated the performance using three metrics:

- **Precision/Recall:** Comparing generated values against ground truth.
- **Coverage:** Percentage of samples for which the model produces any output. For certain attributes (e.g., *subject_character* in t-shirts), low coverage with high precision is expected and desired, as many products naturally do not have these attributes. This metric helps assess whether the model appropriately identifies cases where attribute values should or should not be generated.

Generated values were evaluated through string comparison with ground truth, with LLM-based semantic verification for inconclusive cases. The ground truth included negative labels (Not Applicable, Not Obtainable) to penalize incorrect generations.

## 5 RESULTS

### 5.1 PERFORMANCE OF CASCADEAGENT

We first evaluate the performance of CascadeAgent. Table 1 compares CascadeAgent approach with a baseline method. The baseline uses a single prompt for all attributes where as CascadeAgent uses individual instructions for each attribute through to its Multi-pass Prompt Generation (MPG) framework as well as catalog guidelines and seller preferences through the multi-agent cascade. As an ablation, to understand the value add from MPG vs multi-agent cascade with catalog guidelines and seller preferences we additionally compare with an MPG without the agents.

Table 1: Comparison of CascadeAgent vs. Baseline vs. vanilla MPG

| Base Model | Prompt Type | Precision (%) | Coverage (%) |
|---|---|---|---|
| Mistral NeMo | Baseline | 57.14 | 42.58 |
| Mistral NeMo | MPG | 76.19 | 58.10 |
| Mistral NeMo | CascadeAgent with MPG | 90.21 | 56.09 |
| Claude 3.5 Sonnet | Baseline | 72.73 | 68.47 |
| Claude 3.5 Sonnet | MPG | 87.32 | 86.02 |
| Claude 3.5 Sonnet | CascadeAgent with MPG | 93.55 | 86.49 |

To robustly access our instruction generation ability we used Mistral NeMo and Claude 3.5 Sonnet as generators and evaluated against ground truth human labels, measuring coverage (proportion of attributes filled) and precision.

Results on the 10k Catalog evaluation set show significant improvements from CascadeAgent with MPG for both models. Mistral NeMo's coverage increased from 42.58% to 56.09%, and precision from 57.14% to 90.21%. Claude 3.5 Sonnet saw coverage rise from 68.47% to 86.49% and precision from 72.73% to 93.55%. CascadeAgent improved Mistral NeMo's performance more than Claude 3.5 Sonnet's (+33.07% vs +20.82% in Precision, and +13.51% vs +12.02% in Recall). Notably, CascadeAgent enabled the more cost-effective Mistral NeMo to close the Precision gap with premium Claude 3.5 Sonnet to only 3%. This demonstrates CascadeAgent's ability to make affordable models viable for scalable deployment, while premium models can be used to fulfill coverage gaps.

### 5.2 BOOST IN PA-LEVEL PERFORMANCE

We analyzed precision and coverage improvements on the 2,897 PAs representing high-impact attributes in our catalog. As shown in Figure 2, 58.2% of PAs showed consistent positive impact—improving both metrics or enhancing one without degrading the other—while only 2.4% declined in both. The remaining 39.4% exhibited mixed trends, primarily (35.0%) increased coverage with reduced precision, and occasionally (4.4%) the reverse. These trade-offs often stemmed from hallucinations, where the model defaulted to common values (e.g., *plastic* for a computer mouse) based on catalog priors in the absence of sufficient input context.

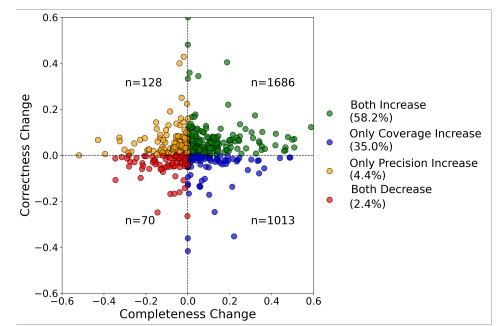

Figure 2: PA specific LLM instructions – coverage and precision changes with quadrants.

### 5.3 OPTIMIZATION WITH TEXTUAL GRADIENTS

We next evaluate continuous optimization of the CascadeAgent over multiple iterations using textual gradients. For this, we used the 1,879 PA high-fidelity set, each with a minimum of

250 ground truth labels. The results demonstrate that 22% (409 PAs) achieved 95% accuracy after the first cascade run. For the remaining 78% (1470 PAs), we employed Algorithm 1 with textual gradient descent and two stopping criteria: reaching 95% train/validation accuracy or completing a maximum of five iterations. This optimization approach proved effective, with all the 1,470 PAs demonstrating substantial performance gains - an average +15% improvement in hold-out test accuracy, comprising +6% in precision and +8% in recall across five iterations (Figure 3). These improvements were all statistically significant (paired $t$-test, $p \leq 1 \times 10^{-10}$).

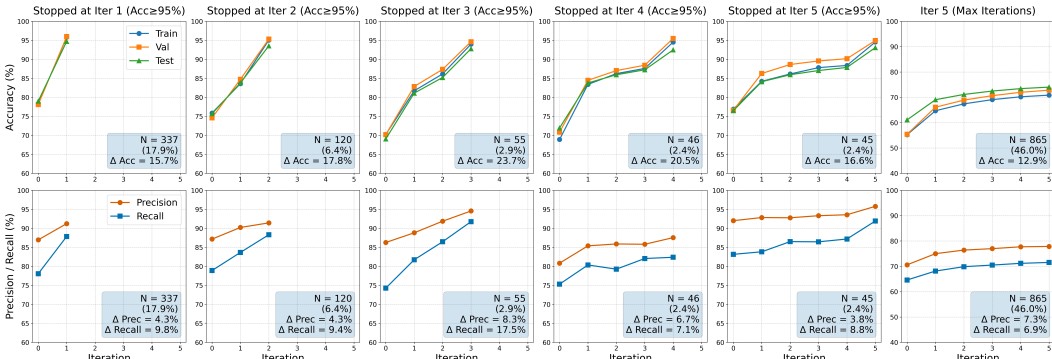

Figure 3: PA specific instructions - Accuracy, Precision/Recall changes over iterations, grouped by stopping criteria.

The optimization process revealed distinct performance patterns: 27.2% of PAs reached the stopping criteria within three iterations, while 4.8% achieved additional performance improvements in the last two iterations. The remaining 46.0% of PAs, though unable to reach the stopping criteria within five iterations, demonstrated substantial improvements during the first three iterations before plateauing(see the last column in Figure 3).

Analysis of cases requiring all five iterations identified two main challenges: image-dependent attributes (40% of remaining cases) and numeric attributes, suggesting opportunities for multimodal models and enhanced mathematical reasoning capabilities.

## 6 RELATED WORKS

Artificial intelligence applications in e-commerce catalog curation have driven significant research across academia and industry (Ghani et al., 2006; Probst et al., 2007; Carmel et al., 2018; Rezk et al., 2019; Zhao et al., 2019; Chen et al., 2019; Cheng et al., 2024). The field has evolved from rule-based systems (Chiticariu et al., 2010; Vandic et al., 2012) to neural architectures, and now to LLMs and multimodal systems.

Early attribute extraction relied on sequence tagging models like OpenTag Zheng et al. (2018) and AdaTag Yan et al. (2021b). With the rise of transformers (Vaswani et al., 2017), question-answering frameworks like MAVE Yang et al. (2022) emerged. While these methods, along with NER (Putthividhya & Hu, 2011; More, 2016; Yan et al., 2021a; Nadeau & Sekine, 2007) and advanced sequence taggers (Xu et al., 2019; Wang et al., 2020), showed progress, they required explicit attribute mentions and complete retraining for new attributes. Recent work in prompt optimization, such as ProTeGi Pryzant et al. (2023), TextGrad Yuksekgonul et al. (2024) and Reflexion Shinn et al. (2023), has demonstrated success through iterative refinement with feedback and self-reflection, but lacks applicability to large-scale e-commerce systems.

Table 2 provides a comprehensive comparison of these approaches, highlighting the evolution of the field and the positioning of our proposed CascadeAgent framework. While recent methods have made progress in individual aspects, they typically require complete retraining when extending to new attributes. In contrast, CascadeAgent's novel multi-agent architecture

Table 2: Comparison of product attribute extraction approaches

| Approach | Re-Train. Required | Implicit Values Inference | Multi-modal | PT/Attr. Scale*** |
|---|---|---|---|---|
| OpenTag (2018) Zheng et al. (2018) | Yes | No | No | Small |
| AdaTag (2021) Yan et al. (2021b) | Yes | No | No | Small |
| MAVE (2022) Yang et al. (2022) | Yes | No | Yes | Medium |
| SAGE (2023) Nikolakopoulos et al. (2023) | Yes | Yes | No | Large |
| LLM-Ensemble (2023) Fang et al. (2024) | No* | Yes | No | Medium |
| DALLA (2023) Zhang et al. (2024) | Yes | No | No | Medium |
| EIVEN (2024) Zou et al. (2024) | Yes | Yes | Yes | Medium |
| ViOC-AG (2024) Gong et al. (2025) | No** | Yes | Yes | Medium |
| MXT (2024) Khandelwal et al. (2023) | Yes | Yes | Yes | Large |
| **CascadeAgent** | **No** | **Yes** | **Yes** | **Large** |

*Requires 2 examples for few-shot learning
**Uses frozen CLIP model with OCR correction
***Small = $\mathcal{O}(10)$; Medium = $\mathcal{O}(1,000)$; Large = $\mathcal{O}(10,000)$

enables seamless extension to new attributes without retraining, while supporting multiple languages and scaling across diverse product categories through its modular framework.

## 7 CONCLUSION

We introduced **CascadeAgent**, a novel multi-agent framework for automating the development of specialized LLM prompts at scale, particularly for e-commerce attribute generation. CascadeAgent leverages **Multi-pass Prompt Generation(MPG)** to modularize tasks into attribute-specific sub-problems, each with a dedicated prompt. A team of five specialized agents then collaboratively creates, evaluates, and refines these prompts through iterative, semantic gradient-based optimization, incorporating domain knowledge.

Empirically, CascadeAgent successfully managed over 27,000 prompts, delivering substantial improvements of +21% to +33% in precision and +12% to +14% in coverage, while optimization via textual gradients resulted in additional gains. Optimization through textual gradients provided additional gains of +6% precision and +8% coverage. Theoretically, we provided a formal analysis demonstrating CascadeAgent's provable convergence towards reduced catalog loss, validating its principled design. This synergy between MPG and the agentic cascade reduces manual effort, enables robust optimization, and allows cost-effective LLMs to achieve high performance.

CascadeAgent's demonstrated scalability and effectiveness highlight its potential for industrial applications requiring nuanced, task-specific LLM instruction. Future work will focus on enhancing agent capabilities and exploring advanced self-reflection mechanisms to further improve performance and adaptability.

## 8 LIMITATIONS

While CascadeAgent demonstrates significant improvements in e-commerce attribute enrichment, several limitations should be considered. Our framework has been validated primarily in e-commerce, with transferability to other domains remaining untested. The system's computational requirements—involving multiple specialized agents—may limit accessibility in resource-constrained environments. Additionally, our approach depends on ground truth data availability, which may be scarce in some domains. Future work could explore reducing computational requirements through more efficient agent orchestration Shinn et al. (2023) and developing semi-supervised approaches Zhou et al. (2023); Chang et al. (2024), leveraging agent-driven data synthesis and/or augmentation Tan et al. (2024) to decrease reliance on labeled data.

## 9 ETHICS STATEMENT

This work focuses on e-commerce catalog enrichment using AI systems. While our system processes product data at scale, we have taken steps to ensure ethical considerations are addressed. Our framework does not process any personally identifiable information or sensitive customer data. The system operates solely on product descriptions and attributes provided by sellers. We have designed CascadeAgent to maintain data quality and accuracy, ensuring that generated attributes fairly represent products without introducing bias or misleading information that could impact consumer decisions. The system's outputs are subject to human oversight and validation to maintain high standards of accuracy and fairness in e-commerce listings.

## 10 REPRODUCIBILITY STATEMENT

To ensure reproducibility of our results, we provide detailed specifications of our experimental setup and methodology. While our implementation code and dataset cannot be released due to company policy, we have thoroughly documented our approach to enable replication. The CascadeAgent framework has been implemented using publicly available LLM API (Claude) and open-source model (Mistral NeMo), with all hyperparameters and configuration settings documented in Section 4.3. The evaluation metrics and methodology are clearly defined in Section 4.4. The experiments can be replicated using standard computing infrastructure as specified in our experimental setup. All results reported in Section 5 are averaged over multiple runs to ensure statistical significance. The theoretical proofs in Section 3 are provided with complete derivations in the appendix B. We believe these detailed specifications allow others to implement and validate our approach, even without access to our proprietary code and data.

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

Appendix

## A  Appendix

Table 3: Benchmarking Evaluation on Enriching Simulated Empty Catalog*

|  | Coverage (%) | Precision (%) |
| --- | --- | --- |
| No LLM Instruction | 45.02 | 81.09 |
| Attribute LLM Instruction | 40.92 | 88.56 |
| PA Specific LLM Instruction | 52.00 | 89.05 |

- For the same 10,000 catalog evaluation set, we masked all catalog attribute values and tasked the model with enriching them using only the product title, bullet_point, and product_description as input data.

Table 4: Benchmarking Evaluation on Enriching Catalog*

|  | Coverage (%) | Precision (%) |
| --- | --- | --- |
| No LLM Instruction | 65.47 | 84.01 |
| Attribute LLM Instruction | 65.18 | 86.20 |
| PA Specific LLM Instruction | 67.29 | 87.94 |

- The evaluation was performed using all attributes as input and with the existing filled catalog.

### A.1  Ablation Studies

We conducted comprehensive ablation studies to analyze the impact of various hyperparameters on attribute generation performance during the iterative prompt optimization. In each study, we modified one parameter while maintaining others at their optimal values. We investigated three key parameters: Top-K selection, number of error examples, and number of minibatches.

#### A.1.1  LLM instruction optimization - Top K

Selecting the optimal **Top-K** instructions at each iteration is critical for maintaining an effective balance between exploration and exploitation.

Figure 4 illustrates the impact of varying the Top-K values (K=1,2,3) across three iterations for 10 PAs. The results show that Top-2 and Top-3 consistently yielded performance improvements across iterations. In contrast, Top-1 showed limited improvement after the first iteration, suggesting convergence to a local optimum.

This behavior indicates that relying solely on the single best-performing instruction (Top-1) constrains the model's ability to explore diverse solutions. Using Top-2 or Top-3 provide a broader spectrum of high-performing instructions, facilitating better exploration of the solution space while maintaining performance quality. This balance between diversity and performance makes Top-2 and Top-3 more effective at achieving sustained improvements and avoiding premature convergence.

#### A.1.2  LLM instruction optimization - Error Examples in Minibatch

Figure 5 analyzes how varying the number of error examples in the flaw summary affects the optimization process.

We evaluated three configurations (1, 3, and 5 error examples) across three iterations. Using a single error example per minibatch resulted in poor convergence and minimal accuracy improvements on the hold-out test dataset. Increasing to 3 or 5 error examples per minibatch significantly enhanced error generalization, leading to substantial accuracy

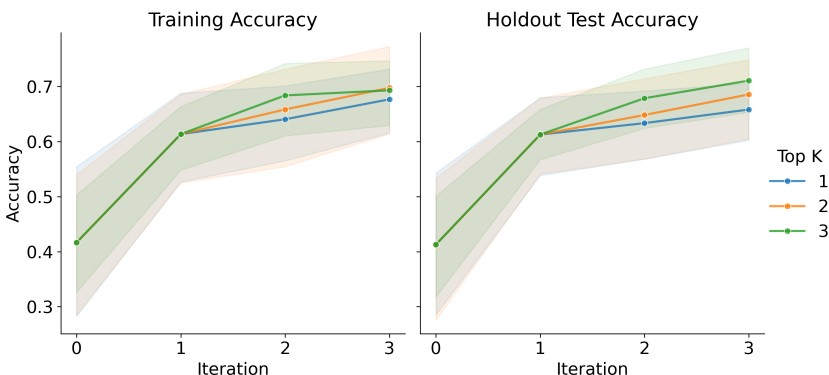

Figure 4: Performance comparison across different Top-K selection strategies

improvements across iterations. This suggests that a larger set of error examples provides more comprehensive guidance for the optimization process.

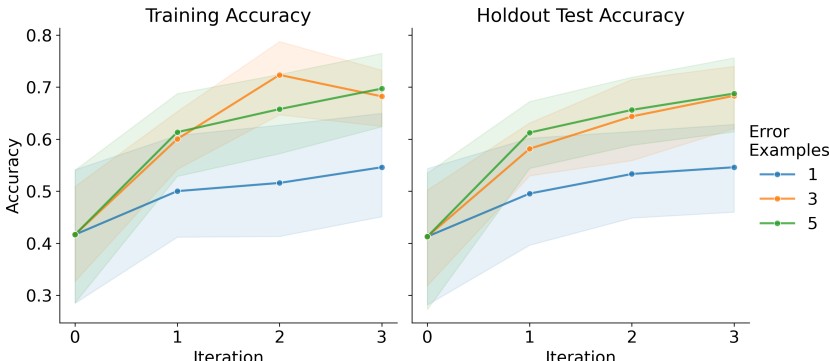

Figure 5: Impact of error example count on model performance and convergence

### A.1.3 LLM INSTRUCTION OPTIMIZATION - MINIBATCH SIZE

Figure 6 examines the effect of varying the number of minibatches during each training iteration.

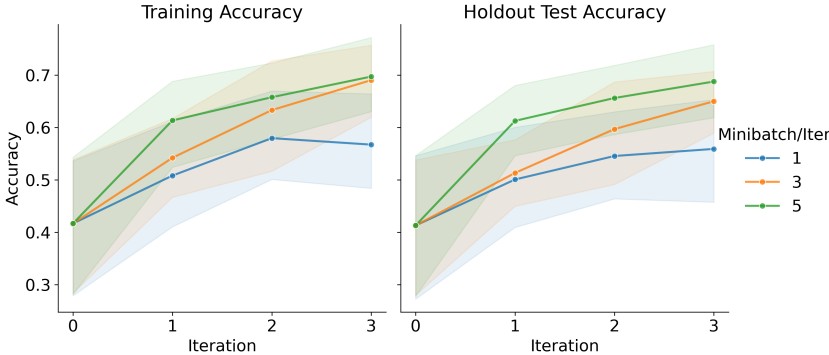

Figure 6: Effect of minibatch count on convergence rate and model performance

We tested three configurations (1, 3, and 5 minibatches) across three iterations. Results show that higher minibatch counts (3 and 5) achieve faster convergence and superior performance

within the first two iterations. Training with a single minibatch per iteration exhibits slower convergence, requiring more iterations to achieve comparable performance improvements.

## A.2 Prompt Structure

Our prompt architecture consists of multiple hierarchical components, as illustrated in Figure 7. The structure follows a systematic organization:

1. **System Role**: We assign the LLM a domain-specific role ("You are a Catalog Expert") to establish appropriate context for catalog enrichment tasks.
2. **Product Context**: We concatenate comprehensive product information including product category, title, description, bullet points, and other available attribute data to provide complete context for attribute generation.
3. **Task Definition**: We specify the task type through a task-specific prompt, indicating whether the objective is to generate a missing attribute value or correct an existing one.
4. **Attribute-Specific Instructions**: We incorporate attribute-related metadata or LLM instructions tailored to the specific product-attribute pair.
5. **Output Schema**: We define the expected output format to ensure consistent, structured responses.

**Baseline Configurations** Our experimental comparison in Table 1 differ in the level of prompt specialization:

- **Baseline (No LLM Instruction)**: Uses the prompt structure shown in the left side of Figure 7 but excludes the attribute metadata module. This represents a universal prompt template applicable to all product-attributes, with only the product details and attribute name varying across instances.
- **MPG (Multi-pass Prompt Generation)**: Incorporates the complete prompt structure shown in the left side of Figure 7, including the attribute metadata module.
- **MPG + CascadeAgent**: Extends MPG by using the Writing Agent to analyze comprehensive product-attribute metadata (including attribute definitions, data types, business logic and etc.) and automatically draft optimized, attribute-specific instructions at scale. These generated instructions are inserted after the task-specific prompt section. The evaluation of MPG + CascadeAgent in Table 1 was performed after the iterative refinement process described next.

**Iterative Refinement Process** The iterative optimization workflow is illustrated in Figure 8. Following the initial cascade run with large-scale instruction generation, the refinement process proceeds as follows:

**1. Generation and Evaluation.** The Prompting Agent constructs complete prompts for each product-attribute pair in the training set. The Generation Agent produces attribute values using these prompts. The Evaluation Agent then assesses output quality: for numeric and hard-enumerated attributes, we use strict synonym matching; for string, boolean, and other unstructured attributes, we use LLM-based evaluation (Mistral NeMo) with a modified prompt structure (similar to the generation prompt, differing only in the task specification) where the task compares generated value $x$ against ground truth $y$.

**2. Error Sampling and Flaw Detection.** We filter all incorrect predictions from the training set and randomly sample 3-5 error cases to form the input for the Flaw Agent. The Flaw Agent receives a system prompt ("Analyze issues with the current LLM instruction by examining errors in the provided samples") along with error examples concatenated as: `<error_ASIN_1><title><description><bullets></error_ASIN_1>` `<error_ASIN_2>...</error_ASIN_2>` ... `<error_ASIN_n>...</error_ASIN_n>`. The Flaw Agent produces a systematic error summary identifying patterns and root causes.

**3. Prompt Rewriting.** The Writing Agent receives the current LLM instruction, the error summary from the Flaw Agent, and the task: "Improve the LLM instruction to avoid

similar errors identified in the error summary." This error sampling → flaw detection → rewriting cycle is repeated 5 times with different random error samples, generating multiple candidate prompt variants. Several key considerations guide this step: First, our training data includes labels where the ground truth is "the value is not obtainable or not applicable." Generations in these cases are considered errors to prevent hallucination. Second, after initial exploration, we added multiple rules for the Writing Agent to follow, including: "Avoid giving default values in the instructions," "The instruction should be generalized to work for any marketplace and language," "Use terminology consistent with eCommerce taxonomy and the specific product category," and "Be concise, within 1-3 sentences and a maximum of 200 words." These rules regulate the quality and consistency of LLM instructions.

**4. Re-evaluation and Selection.** All candidate prompts are used to regenerate attribute values via the Generation Agent. The Evaluation Agent assesses performance of each variant. Top-K selection identifies the best-performing prompts for the next iteration.

This iterative process continues until convergence criteria are met or the maximum iteration count is reached, systematically refining prompts based on observed failure patterns.

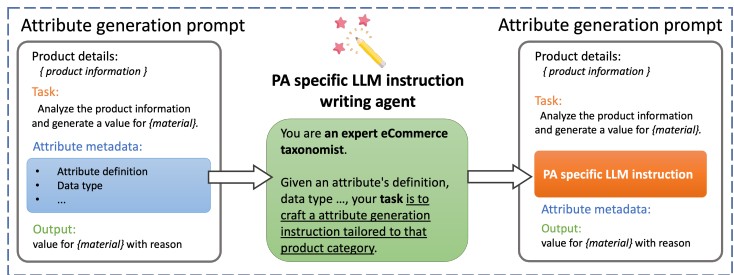

Figure 7: Prompt generation process by writing agent.

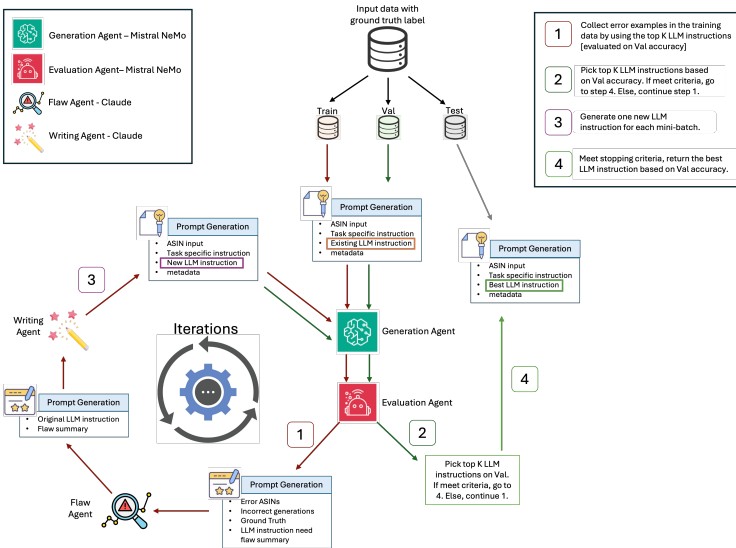

Figure 8: Prompt iterative refinement process.

### A.3 EXAMPLE OPTIMIZED INSTRUCTIONS

We present three representative examples demonstrating how CascadeAgent's iterative refinement process transforms initial attribute definitions into optimized, context-aware instructions that significantly improve performance.

**Example 1: BACKPACK – strap_type**

**Initial instruction:** "This attribute indicates the type of strap that the item has."

**Optimized instruction (after iterative refinement):** "The `strap_type` attribute for backpacks refers to the material, design, and construction of the shoulder straps used to secure the backpack to the wearer's body. Common strap types for backpacks include padded mesh straps for breathability and comfort, reinforced nylon straps for durability, or adjustable sternum/chest straps for added stability during activities like hiking or cycling. The `strap_type` attribute helps consumers understand the level of comfort, support, and functionality provided by the backpack's strap system."

**Performance improvement:** Recall improved from 63% to 80%; precision improved from 85% to 94%.

**Example 2: FACIAL_TISSUE – item_form**

**Initial instruction:** None.

**Optimized instruction (after iterative refinement):** "For the FACIAL_TISSUE product category, the `item_form` attribute represents the physical form or structure in which the facial tissue product is available. The valid value for this attribute should be 'Sheets' as facial tissues are typically sold as flat, individual sheets packaged together in a box or pouch. Avoid assigning values like 'cylindrical', 'bag', or 'rectangular' which are not applicable to facial tissue products. If the product details do not explicitly state the item form, provide the value as '[NO]'. If the attribute is not relevant for a specific facial tissue product listing, use '[NA]' as the value."

**Performance improvement:** Recall improved from 83% to 85%; precision improved from 54% to 89%.

**Example 3: CANDLE – style**

**Initial instruction:** None.

**After initial cascade run:** "For candles, the 'style' attribute refers to the distinctive visual design and aesthetic appeal of the candle. It encompasses elements such as the shape, color, pattern, or surface embellishments that create a particular look or ambiance. Common style values for candles include 'minimalist,' 'vintage,' 'ornate,' 'rustic,' or 'modern,' each evoking a different sensory experience and complementing various decor themes or personal preferences."

**After iterative refinement:** "For the `style` attribute of candles, specify the predominant aesthetic or design theme based on shape, materials, ornamentation, and intended decor purposes. For plain, unscented candles without discernible stylistic elements, use 'Basic' or 'Undecorated'. Separate size descriptors like 'Compact', 'Travel Size', or dimensions from stylistic qualifiers. Valid value conventions:

- Use 'Basic', 'Undecorated', 'Plain', or simply omit this attribute for unscented candles lacking stylistic design elements.
- '[Cultural Style]' (e.g., 'Mediterranean', 'Chinese') for designs inspired by regional artistic traditions or motifs.
- 'Traditional' or 'Classic' for timeless, vintage aesthetics or nostalgic designs reminiscent of bygone eras.
- 'Modern', 'Contemporary', or 'Minimalist' for current, clean-lined styles following minimalist design principles.
- '[Material]' (e.g., 'Rustic', 'Tropical') when the primary material imparts a distinct natural or earthy style.
- 'Custom', 'Novelty', 'Decorative', or 'Occasion' for unconventional shapes, artistic designs, or occasion-specific candle products.

If the style cannot be deduced, give '[NO]'. If not applicable, give '[NA]'."

**Performance improvement:** Recall remained stable at 83%; precision improved from 52% to 76% after initial cascade, then to 85% after iterative refinement.

**Key Observations**  These examples illustrate several patterns in how CascadeAgent optimizes instructions:

1. **Domain-specific contextualization**: The optimized instructions incorporate product category knowledge (e.g., backpack strap functionality, facial tissue form factors).
2. **Explicit value guidance**: Instructions specify valid and invalid values, reducing hallucination and improving precision.
3. **Structured decision rules**: Complex attributes like candle style receive hierarchical guidance with clear conventions for edge cases.
4. **Iterative refinement value**: Example 3 demonstrates how multiple refinement iterations progressively improve instruction quality, with precision gains of +24% (initial cascade) and an additional +9% (iterative refinement).

# B  PROOFS

## B.1  PROPOSITION 1: FORMAL STATEMENT AND PROOF

**Proposition 1** (Empirical Top-$K$ selection guarantees monotone improvement). *Fix an iteration $t$. Let $\mathcal{P}_t \subseteq \Pi$ be the current pool of instructions and define*

$$\pi_t^\star := \arg\min_{\pi \in \mathcal{P}_t} \mathcal{L}(\pi), \qquad \Delta_t := \min_{\pi \neq \pi_t^\star} \big[\mathcal{L}(\pi) - \mathcal{L}(\pi_t^\star)\big] > 0.$$

*(Loss range)  Because $w_{val} + w_{omis} + w_{commis} = 1$ we have $0 \leq \mathcal{L}(\pi) \leq 1$ for every $\pi \in \mathcal{P}_t$.*

*(Evaluation batch)  Draw a fresh i.i.d. sample $\mathcal{D}_t = \{(x_i, y_i)\}_{i=1}^n$ and set the empirical loss*

$$\widehat{\mathcal{L}}_t(\pi) := \frac{1}{n} \sum_{i=1}^n \mathbf{1}\big[f_\theta(x_i, \pi) \neq y_i\big], \qquad \pi \in \mathcal{P}_t.$$

*Choose confidence level $\vartheta \in (0, 1)$ and any*

$$n \geq \Big\lceil \frac{2}{\Delta_t^2} \log \frac{2|\mathcal{P}_t|}{\vartheta} \Big\rceil.$$

*Let $\widehat{\mathcal{P}}_{t,K}$ be the empirical Top-$K$ instructions computed from $\widehat{\mathcal{L}}_t(\pi)$ using a deterministic tie-break order, and set $\mathcal{P}_{t+1} := \mathcal{P}_t \cup \widehat{\mathcal{P}}_{t,K}$ (for any $K \geq 1$).*

*Then, with probability at least $1 - \vartheta$,*

*(i) **True optimum retained:** $\pi_t^\star \in \widehat{\mathcal{P}}_{t,K}$.*

*(ii) **Monotone improvement:** $\min\limits_{\pi \in \mathcal{P}_{t+1}} \mathcal{L}(\pi) = \mathcal{L}(\pi_t^\star) \leq \min\limits_{\pi \in \mathcal{P}_t} \mathcal{L}(\pi).$*

*Proof. (Step 1). A uniform concentration event.* For one fixed prompt $\pi \in \mathcal{P}_t$ the random variable $\widehat{\mathcal{L}}_t(\pi)$ is the empirical mean of $n$ independent Bernoulli indicators $\mathbf{1}[f_\theta(x_i, \pi) \neq y_i]$. Because each indicator lies in $[0, 1]$, Hoeffding's inequality states that for any $\varepsilon > 0$

$$\Pr\Big[\big|\widehat{\mathcal{L}}_t(\pi) - \mathcal{L}(\pi)\big| \geq \varepsilon\Big] \leq 2 \exp\big(-2n\varepsilon^2\big). \tag{H}$$

We would like this deviation bound to hold *simultaneously* for every $\pi \in \mathcal{P}_t$. Set

$$\varepsilon := \sqrt{\frac{\log\big(2|\mathcal{P}_t|/\vartheta\big)}{2n}}.$$

With this choice the right–hand side of (H) equals $\vartheta/|\mathcal{P}_t|$. A union bound over all $|\mathcal{P}_t|$ prompts therefore yields the event

$$\mathcal{E}_t := \left\{ \left|\widehat{\mathcal{L}}_t(\pi) - \mathcal{L}(\pi)\right| < \varepsilon \quad \forall \pi \in \mathcal{P}_t \right\},$$

with probability $\Pr[\mathcal{E}_t] \geq 1 - \vartheta$. This event says "*all* empirical losses are within $\varepsilon$ of their true values."

*(Step 2). Empirical ordering mirrors true ordering.* Fix any competitor $\pi \neq \pi_t^\star$. By definition of the true gap $\Delta_t = \mathcal{L}(\pi) - \mathcal{L}(\pi_t^\star)$ we have

$$\mathcal{L}(\pi) - \mathcal{L}(\pi_t^\star) = \Delta_t > 0. \tag{1}$$

On the concentration event $\mathcal{E}_t$ we add and subtract at most $\varepsilon$ to each loss, so

$$\widehat{\mathcal{L}}_t(\pi) - \widehat{\mathcal{L}}_t(\pi_t^\star) \geq \Delta_t - 2\varepsilon. \tag{2}$$

The sample size $n$ was chosen so that $2\varepsilon = \Delta_t \sqrt{\frac{2\log(2|\mathcal{P}_t|/\vartheta)}{n}} \leq \Delta_t$; indeed by the proposition's hypothesis $n \geq \frac{2}{\Delta_t^2} \log \frac{2|\mathcal{P}_t|}{\vartheta}$. Therefore the right-hand side of (2) is $\geq 0$, and in fact $> 0$ because we rounded $n$ *up*. Hence on $\mathcal{E}_t$ every competitor $\pi$ has strictly larger empirical loss than $\pi_t^\star$.

*(Step 3). Inclusion of the optimal prompt in Top-K.* Because of the strict inequality in step 2 and because the tie-break rule is deterministic, the empirical ranking places $\pi_t^\star$ *first*. Consequently it is included in the Top-K set $\widehat{\mathcal{P}}_{t,K}$ for any $K \geq 1$. This establishes part (i) of the proposition.

*(Step 4). Monotone improvement of the best true loss.* Define the next pool $\mathcal{P}_{t+1} := \mathcal{P}_t \cup \widehat{\mathcal{P}}_{t,K}$. Since $\pi_t^\star \in \widehat{\mathcal{P}}_{t,K}$ by part (i), it certainly belongs to $\mathcal{P}_{t+1}$. Therefore

$$\min_{\pi \in \mathcal{P}_{t+1}} \mathcal{L}(\pi) \leq \mathcal{L}(\pi_t^\star) = \min_{\pi \in \mathcal{P}_t} \mathcal{L}(\pi).$$

(The inequality could be strict if an *even better* prompt were added; equality holds at minimum.) This proves part (ii).

*(Step 5). Probability statement.* Steps 2–4 hold on the event $\mathcal{E}_t$, whose probability we bounded below by $1 - \vartheta$ in step 1. Hence parts (i)–(ii) both hold with at least that probability, and the proof is complete. □

## B.2 Proposition 2: Formal Statement and Proof

**Proposition 2** (Geometric loss decay under marginal churn)**.** *Let the state at iteration $t$ be $s_t$, leading to a prediction $\hat{Y}_t$ for a ground-truth value $Y$. Define the per-example error type indicators:*

- $E_{val,t} := \mathbf{1}\{\hat{Y}_t \neq Y \wedge \hat{Y}_t \neq blank \wedge Y \neq blank\}$ *(Incorrect Value)*
- $E_{omis,t} := \mathbf{1}\{\hat{Y}_t = blank \wedge Y \neq blank\}$ *(Omission Error)*
- $E_{commis,t} := \mathbf{1}\{\hat{Y}_t \neq blank \wedge Y = blank\}$ *(Commission Error)*

*These error types are mutually exclusive for a single prediction. The overall binary error indicator is $E_t := \mathbf{1}\{E_{val,t} = 1 \vee E_{omis,t} = 1 \vee E_{commis,t} = 1\}$. The per-example catalog loss at iteration $t$ is:*

$$\mathcal{L}_t := w_{val}E_{val,t} + w_{omis}E_{omis,t} + w_{commis}E_{commis,t},$$

*where $0 < w_{val}, w_{omis}, w_{commis} < 1$ such that $w_{val} + w_{omis} + w_{commis} = 1$. Define the constants:*

$$\kappa_{\min} := \min(w_{val}, w_{omis}, w_{commis}), \quad \kappa_{\max} := \max(w_{val}, w_{omis}, w_{commis}), \quad r := \frac{\kappa_{\max}}{\kappa_{\min}} \geq 1.$$

*Note that if $E_t = 1$, then $\kappa_{\min} \leq \mathcal{L}_t \leq \kappa_{\max}$. If $E_t = 0$, then $\mathcal{L}_t = 0$. Thus, $\kappa_{\min}E_t \leq \mathcal{L}_t \leq \kappa_{\max}E_t$ holds for all outcomes.*

Assume the following **marginal-churn dynamics** hold for the binary error indicator $E_t$, with fixed probabilities $p_{\text{fix}}, p_{\text{break}} \in [0,1]$:

$$\Pr[E_{t+1} = 0 \mid E_t = 1] \;=\; p_{\text{fix}}, \qquad \Pr[E_{t+1} = 1 \mid E_t = 0] \;=\; p_{\text{break}}.$$

Assume a beneficial rewriting process, $p_{\text{fix}} > p_{\text{break}} \geq 0$. Also assume $p_{\text{fix}} + p_{\text{break}} \in (0,1]$ to ensure the standard contraction factor definition. Define the contraction factors:

$$\rho := p_{\text{fix}} + p_{\text{break}} \in (0,1],$$
$$\lambda := 1 - \rho = 1 - (p_{\text{fix}} + p_{\text{break}}) \in [0,1).$$

Set the effective contraction rate $\phi := r\lambda$. We require $\phi \in [0,1)$ for convergence to the stated floor.

Then the following two bounds hold:

   (i) **One-step bound:** $\mathbb{E}\big[\mathcal{L}_{t+1}\big] \;\leq\; \phi\,\mathbb{E}\big[\mathcal{L}_t\big] \;+\; \kappa_{\max}\,p_{\text{break}}$

   (ii) **Multi-step bound:** For any horizon $T \geq 0$,

$$\mathbb{E}\big[\mathcal{L}_T\big] \;\leq\; \phi^T \mathbb{E}\big[\mathcal{L}_0\big] + \kappa_{\max}\,p_{\text{break}}\,\frac{1 - \phi^T}{1 - \phi}.$$

Consequently, if $\phi < 1$, the expected loss converges towards an asymptotic floor:

$$\lim_{T \to \infty} \mathbb{E}[\mathcal{L}_T] \;\leq\; \frac{\kappa_{\max}\,p_{\text{break}}}{1 - \phi}.$$

*Proof.* Step 1: Bounding the loss in terms of the binary error indicator $E_t$. As established in the proposition statement: If $E_t = 1$, an error occurred, and the loss $\mathcal{L}_t$ is one of $w_{\text{val}}, w_{\text{omis}}, w_{\text{commis}}$. Thus, $\kappa_{\min} \leq \mathcal{L}_t \leq \kappa_{\max}$. If $E_t = 0$, no error occurred (the prediction was correct, including predicting blank for blank), and $\mathcal{L}_t = 0$. These two cases can be summarized by the inequality:

$$\kappa_{\min} E_t \;\leq\; \mathcal{L}_t \;\leq\; \kappa_{\max} E_t. \tag{A}$$

This implies $E_t \leq \mathcal{L}_t / \kappa_{\min}$ (if $\kappa_{\min} > 0$, which it is by definition of weights) and $\mathcal{L}_t / \kappa_{\max} \leq E_t$. Also, $\mathbb{E}[E_t] \leq \mathbb{E}[\mathcal{L}_t] / \kappa_{\min}$ and $\mathbb{E}[\mathcal{L}_t] \leq \kappa_{\max} \mathbb{E}[E_t]$.

*Step 2: Expected value of the next error indicator $E_{t+1}$.* By definition of expected value and the law of total probability, using the marginal-churn dynamics for $E_t$:

$$\begin{aligned}
\mathbb{E}[E_{t+1}] &= \Pr(E_{t+1} = 1) \\
&= \Pr(E_{t+1} = 1 \mid E_t = 1)\Pr(E_t = 1) + \Pr(E_{t+1} = 1 \mid E_t = 0)\Pr(E_t = 0) \\
&= (1 - p_{\text{fix}})\mathbb{E}[E_t] + p_{\text{break}}(1 - \mathbb{E}[E_t]) \\
&= (1 - p_{\text{fix}} - p_{\text{break}})\mathbb{E}[E_t] + p_{\text{break}} \\
&= \lambda\,\mathbb{E}[E_t] + p_{\text{break}}. \tag{B}
\end{aligned}$$

Here, $\lambda = 1 - (p_{\text{fix}} + p_{\text{break}})$. The assumption $p_{\text{fix}} + p_{\text{break}} \in (0,1]$ ensures that $\lambda \in [0,1)$. The condition $p_{\text{fix}} > p_{\text{break}} \geq 0$ ensures that $\lambda < 1$ if $p_{\text{fix}} + p_{\text{break}} > 0$.

*Step 3: Deriving the one-step loss bound (i).* Using the right-hand inequality of (A) for $\mathcal{L}_{t+1}$ and taking expectations:

$$\mathbb{E}[\mathcal{L}_{t+1}] \leq \kappa_{\max}\mathbb{E}[E_{t+1}].$$

Substituting (B) into this:

$$\mathbb{E}[\mathcal{L}_{t+1}] \leq \kappa_{\max}(\lambda\mathbb{E}[E_t] + p_{\text{break}}).$$

Now, using the implication from the left-hand inequality of (A), $\mathbb{E}[E_t] \leq \mathbb{E}[\mathcal{L}_t]/\kappa_{\min}$:

$$\mathbb{E}[\mathcal{L}_{t+1}] \leq \kappa_{\max}\left(\lambda\frac{\mathbb{E}[\mathcal{L}_t]}{\kappa_{\min}} + p_{\text{break}}\right).$$

$$\mathbb{E}[\mathcal{L}_{t+1}] \leq \lambda\frac{\kappa_{\max}}{\kappa_{\min}}\mathbb{E}[\mathcal{L}_t] + \kappa_{\max}p_{\text{break}}.$$

Since $r = \kappa_{\max}/\kappa_{\min}$ and $\phi = r\lambda$, this becomes:

$$\mathbb{E}[\mathcal{L}_{t+1}] \leq \phi\,\mathbb{E}[\mathcal{L}_t] + \kappa_{\max}p_{\text{break}}.$$

This proves part (i).

*Step 4: Deriving the multi-step bound (ii).* Let $x_t = \mathbb{E}[\mathcal{L}_t]$ and $c = \kappa_{\max}p_{\text{break}}$. The recurrence from part (i) is $x_{t+1} \leq \phi x_t + c$. We unroll this recurrence:

$$x_1 \leq \phi x_0 + c$$
$$x_2 \leq \phi x_1 + c \leq \phi(\phi x_0 + c) + c = \phi^2 x_0 + \phi c + c$$
$$x_3 \leq \phi x_2 + c \leq \phi(\phi^2 x_0 + \phi c + c) + c = \phi^3 x_0 + \phi^2 c + \phi c + c$$
$$\vdots$$
$$x_T \leq \phi^T x_0 + c(\phi^{T-1} + \phi^{T-2} + \cdots + \phi^1 + \phi^0)$$
$$x_T \leq \phi^T x_0 + c\sum_{k=0}^{T-1}\phi^k.$$

The sum is a geometric series. We have required $\phi \in [0,1)$, so $\phi \neq 1$. Thus, $\sum_{k=0}^{T-1}\phi^k = \frac{1-\phi^T}{1-\phi}$.

$$\mathbb{E}[\mathcal{L}_T] \leq \phi^T\mathbb{E}[\mathcal{L}_0] + \kappa_{\max}p_{\text{break}}\frac{1-\phi^T}{1-\phi}.$$

This proves part (ii).

*Step 5: Asymptotic behavior.* As $T \to \infty$, if $\phi \in [0,1)$, then $\phi^T \to 0$. Therefore,

$$\lim_{T\to\infty}\mathbb{E}[\mathcal{L}_T] \leq \kappa_{\max}p_{\text{break}}\frac{1}{1-\phi} = \frac{\kappa_{\max}p_{\text{break}}}{1-r\lambda}.$$

This completes the proof. $\qquad\square$

### B.3 Theorem 1: Formal Statement and Proof

**Theorem 1** (Loss and value improvement of $\varpi_{\text{TG}}$ under corrected definitions). *Fix a single catalog attribute and let $s_t \in \Pi$ be the prompt (instruction) active at iteration $t \geq 0$.*

**Assumptions.**

(a) Estimator step. *At each iteration, the empirical Top-K selector of Proposition 1 (from Appendix B.1) is run. Denote by $\mathcal{E}_t := \{$true best prompt from the current pool is retained by Top-K selection$\}$ and let $\vartheta_{\text{est}} := \Pr[\mathcal{E}_t^c]$ be its failure probability.*

(b) Catalog dynamics (marginal churn). *Per-example binary error indicators $E_t$ (indicating any true error: $E_{val,t}$, $E_{omis,t}$, or $E_{commis,t}$) obey the model of Proposition 2 with fixed parameters $p_{\text{fix}} > p_{\text{break}} \geq 0$, and $p_{\text{fix}} + p_{\text{break}} \in (0,1]$. The catalog loss $\mathcal{L}(s_t)$ is defined as in Proposition 2, with $0 < \kappa_{\min} \leq \kappa_{\max}$. Define the contraction factors:*

$$\lambda := 1 - (p_{\text{fix}} + p_{\text{break}}) \in [0,1),$$
$$r := \kappa_{\max}/\kappa_{\min} \geq 1,$$
$$\phi := r\lambda.$$

*We assume the strict effective contraction condition $\phi \in [0,1)$.*

**Claims.** *For every state $s_t$ and every horizon $T \geq 0$:*

(i) **One-step loss recursion (conditional on $s_t$):** *The expected loss $\mathcal{L}(s_{t+1})$ of the prompt chosen for the next step, conditional on the current prompt $s_t$ (which has loss $\mathcal{L}(s_t)$), satisfies:*

$$\mathbb{E}\big[\mathcal{L}(s_{t+1}) \mid s_t\big] \leq \phi\,\mathcal{L}(s_t) + \kappa_{\max}\,p_{\text{break}} + \vartheta_{\text{est}}. \tag{A'}$$

(ii) **T-step horizon bound (unconditional expectation):** *Let $\mathcal{L}_0 = \mathcal{L}(s_0)$ be the loss of the initial prompt.*

$$\mathbb{E}\big[\mathcal{L}(s_T)\big] \;\leq\; \phi^T \,\mathcal{L}_0 + \frac{\kappa_{\max}\, p_{\text{break}} + \vartheta_{\text{est}}}{1 - \phi} \,\big(1 - \phi^T\big). \tag{B'}$$

(iii) **Value-function improvement:** *Set the asymptotic loss floor*

$$\mathcal{L}_\infty^\star := \frac{\kappa_{\max}\, p_{\text{break}} + \vartheta_{\text{est}}}{1 - \phi}.$$

*Define the excess loss $\widetilde{\mathcal{L}}(s) := \mathcal{L}(s) - \mathcal{L}_\infty^\star$ and the shifted reward $r'(s) := -\widetilde{\mathcal{L}}(s)$. For any discount factor $\gamma \in (0,1)$, the policy $\varpi_{\text{TG}}$ satisfies:*

$$\mathbb{E}\big[V^{\varpi_{\text{TG}}}(s_{t+1}) - V^{\varpi_{\text{TG}}}(s_t) \mid s_t\big] \;\geq\; \frac{1-\phi}{1-\gamma\phi}\,\big[\mathcal{L}(s_t) - \mathcal{L}_\infty^\star\big]^+, \tag{C'}$$

*where $[x]^+ = \max(0, x)$.*

*Proof.* Throughout this proof, $\mathcal{L}_t$ refers to $\mathcal{L}(s_t)$, the true loss of the prompt at state $s_t$.

*Part (i): One-step loss recursion.* The expectation of $\mathcal{L}(s_{t+1})$ conditioned on $s_t$ is expanded by conditioning on the success or failure of the Top-K estimation step $\mathcal{E}_t$:

$$\begin{aligned}
\mathbb{E}[\mathcal{L}(s_{t+1}) \mid s_t] &= \mathbb{E}[\mathcal{L}(s_{t+1}) \mid s_t, \mathcal{E}_t]\Pr(\mathcal{E}_t) + \mathbb{E}[\mathcal{L}(s_{t+1}) \mid s_t, \mathcal{E}_t^c]\Pr(\mathcal{E}_t^c) \\
&= \mathbb{E}[\mathcal{L}(s_{t+1}) \mid s_t, \mathcal{E}_t](1 - \vartheta_{\text{est}}) + \mathbb{E}[\mathcal{L}(s_{t+1}) \mid s_t, \mathcal{E}_t^c]\vartheta_{\text{est}}.
\end{aligned}$$

On event $\mathcal{E}_t$, the Top-K selection manages to retain the best prompt. The expected loss of a prompt $\pi'$ generated from $s_t$ via rewriting, before Top-K selection from the wider pool, is bounded by Proposition 2(i): $\mathbb{E}_{\pi'}[\mathcal{L}(\pi')] \leq \phi\,\mathcal{L}_t + \kappa_{\max}\, p_{\text{break}}$. Since $s_{t+1}$ is chosen by Top-K, its loss will be at most this value if a new prompt is chosen, or potentially lower if an existing better prompt is kept. Thus,

$$\mathbb{E}[\mathcal{L}(s_{t+1}) \mid s_t, \mathcal{E}_t] \leq \phi\,\mathcal{L}_t + \kappa_{\max}\, p_{\text{break}}.$$

On event $\mathcal{E}_t^c$ (Top-K estimation fails to identify the true best among candidates), we use the general upper bound $\mathcal{L}(s_{t+1}) \leq 1$ (assuming losses are normalized or $\kappa_{\max} \leq 1$). If not normalized, the bound is $\kappa_{\max}$. For consistency with the $\vartheta_{\text{est}}$ term, we use 1 as a simple upper bound representing a high-loss state.

$$\mathbb{E}[\mathcal{L}(s_{t+1}) \mid s_t] \leq (1 - \vartheta_{\text{est}})(\phi\,\mathcal{L}_t + \kappa_{\max}\, p_{\text{break}}) + \vartheta_{\text{est}} \cdot 1.$$

Since $\phi\,\mathcal{L}_t + \kappa_{\max}\, p_{\text{break}} \geq 0$, the term $-\vartheta_{\text{est}}(\phi\,\mathcal{L}_t + \kappa_{\max}\, p_{\text{break}})$ is non-positive. Therefore,

$$\begin{aligned}
\mathbb{E}[\mathcal{L}(s_{t+1}) \mid s_t] &\leq \phi\,\mathcal{L}_t + \kappa_{\max}\, p_{\text{break}} - \vartheta_{\text{est}}(\phi\,\mathcal{L}_t + \kappa_{\max}\, p_{\text{break}}) + \vartheta_{\text{est}} \\
&\leq \phi\,\mathcal{L}_t + \kappa_{\max}\, p_{\text{break}} + \vartheta_{\text{est}}.
\end{aligned}$$

and (A') is established.

*Part (ii): T-step horizon bound.* Let $x_t = \mathbb{E}[\mathcal{L}_t]$. Taking the total expectation of (A'):

$$\mathbb{E}[\mathbb{E}[\mathcal{L}(s_{t+1}) \mid s_t]] \leq \mathbb{E}[\phi\,\mathcal{L}_t + \kappa_{\max}\, p_{\text{break}} + \vartheta_{\text{est}}].$$

$$x_{t+1} \leq \phi\, x_t + (\kappa_{\max}\, p_{\text{break}} + \vartheta_{\text{est}}).$$

Let $C := \kappa_{\max}\, p_{\text{break}} + \vartheta_{\text{est}}$. Unrolling this recurrence (as in the proof of Proposition 2(ii)), yields:

$$x_T \leq \phi^T x_0 + C \sum_{k=0}^{T-1} \phi^k = \phi^T x_0 + C\frac{1 - \phi^T}{1 - \phi}.$$

Substituting $x_T = \mathbb{E}[\mathcal{L}_T]$ and $x_0 = \mathcal{L}_0$, and $C$, gives:

$$\mathbb{E}[\mathcal{L}_T] \leq \phi^T \mathcal{L}_0 + (\kappa_{\max}\, p_{\text{break}} + \vartheta_{\text{est}})\frac{1 - \phi^T}{1 - \phi}.$$

which proves (B').

*Part (iii): Value-function improvement.* Recall $\mathcal{L}_\infty^\star = (\kappa_{\max} p_{\text{break}} + \vartheta_{\text{est}})/(1 - \phi)$ and $\widetilde{\mathcal{L}}(s) = \mathcal{L}(s) - \mathcal{L}_\infty^\star$. From (A'), we have

$$\mathbb{E}[\mathcal{L}(s_{t+1}) \mid s_t] \leq \phi\, \mathcal{L}_t + \kappa_{\max}\, p_{\text{break}} + \vartheta_{\text{est}}.$$

Subtracting $\mathcal{L}_\infty^\star$ from both sides:

$$\mathbb{E}[\mathcal{L}(s_{t+1}) \mid s_t] - \mathcal{L}_\infty^\star \leq \phi\, \mathcal{L}_t + \kappa_{\max}\, p_{\text{break}} + \vartheta_{\text{est}} - \mathcal{L}_\infty^\star$$

$$\mathbb{E}[\widetilde{\mathcal{L}}(s_{t+1}) \mid s_t] \leq \phi\, \mathcal{L}_t + \mathcal{L}_\infty^\star(1 - \phi) - \mathcal{L}_\infty^\star \quad (\text{since } \kappa_{\max} p_{\text{break}} + \vartheta_{\text{est}} = \mathcal{L}_\infty^\star(1 - \phi))$$

$$= \phi\, \mathcal{L}_t - \phi\, \mathcal{L}_\infty^\star = \phi\, \widetilde{\mathcal{L}}(s_t). \quad (\dagger)$$

The inequality $\mathbb{E}[\widetilde{\mathcal{L}}(s_{t+1}) \mid s_t] \leq \phi\, \widetilde{\mathcal{L}}(s_t)$ shows that the expected excess loss contracts by $\phi$ at each step. Let $V'(s)$ be the value function associated with the shifted reward $r'(s) = -\widetilde{\mathcal{L}}(s)$. Then

$$V'(s_t) = \mathbb{E}_{\varpi_{\text{TG}}}\left[\sum_{k=0}^{\infty} \gamma^k r'(s_{t+k}) \mid s_t\right]$$

$$= \mathbb{E}_{\varpi_{\text{TG}}}\left[\sum_{k=0}^{\infty} \gamma^k (-\widetilde{\mathcal{L}}(s_{t+k})) \mid s_t\right].$$

Using iterated expectations and the contraction ($\dagger$), $\mathbb{E}[\widetilde{\mathcal{L}}(s_{t+k}) \mid s_t] \leq \phi^k \widetilde{\mathcal{L}}(s_t)$. Thus,

$$V'(s_t) \geq -\sum_{k=0}^{\infty} \gamma^k \phi^k \widetilde{\mathcal{L}}(s_t) = -\frac{1}{1 - \gamma\phi} \widetilde{\mathcal{L}}(s_t).$$

The value function $V^{\varpi_{\text{TG}}}(s_t)$ for the original reward $r(s) = -\mathcal{L}(s)$ is related to $V'(s_t)$ by:

$$V^{\varpi_{\text{TG}}}(s_t) = V'(s_t) - \sum_{k=0}^{\infty} \gamma^k \mathcal{L}_\infty^\star = V'(s_t) - \frac{\mathcal{L}_\infty^\star}{1 - \gamma}.$$

Therefore,

$$\mathbb{E}[V^{\varpi_{\text{TG}}}(s_{t+1}) \mid s_t] - V^{\varpi_{\text{TG}}}(s_t) = \mathbb{E}[V'(s_{t+1}) \mid s_t] - V'(s_t).$$

From the Bellman equation for $V'(s_t)$:

$$V'(s_t) = -\widetilde{\mathcal{L}}(s_t) + \gamma\mathbb{E}[V'(s_{t+1}) \mid s_t].$$

So,

$$\mathbb{E}[V'(s_{t+1}) \mid s_t] - V'(s_t) = \frac{1 - \gamma}{\gamma} V'(s_t) + \frac{1}{\gamma} \widetilde{\mathcal{L}}(s_t).$$

Substituting the bound for $V'(s_t)$:

$$\mathbb{E}[V'(s_{t+1}) \mid s_t] - V'(s_t) \geq \frac{1 - \gamma}{\gamma}\left(-\frac{1}{1 - \gamma\phi}\widetilde{\mathcal{L}}(s_t)\right) + \frac{1}{\gamma}\widetilde{\mathcal{L}}(s_t)$$

$$= \frac{\widetilde{\mathcal{L}}(s_t)}{\gamma}\left(1 - \frac{1 - \gamma}{1 - \gamma\phi}\right) = \frac{\widetilde{\mathcal{L}}(s_t)}{\gamma} \frac{1 - \gamma\phi - (1 - \gamma)}{1 - \gamma\phi}$$

$$= \frac{\widetilde{\mathcal{L}}(s_t)}{\gamma} \frac{\gamma - \gamma\phi}{1 - \gamma\phi} = \frac{1 - \phi}{1 - \gamma\phi}\widetilde{\mathcal{L}}(s_t).$$

Since $[\widetilde{\mathcal{L}}(s_t)]^+ \leq \widetilde{\mathcal{L}}(s_t)$, replacing $\widetilde{\mathcal{L}}(s_t)$ by its positive part preserves the inequality[1], which yields (C'). $\qquad\square$

---

[1]The notation $[x]^+ := \max(0, x)$ is included for clarity: it forces the right-hand side of (C') to be *non-negative*. Hence the inequality states, in plain words, that whenever the current loss sits *above* the asymptotic floor $\mathcal{L}_\infty^\star$, the expected value increases by at least a fixed fraction of that excess; if the loss is already at or below the floor, the bound becomes 0 and the theorem makes no further claim.

