# OpenReview forum: "Prompt Engineering at Scale: Provably Effective Multi-Agent Cascades for Attribute Generation in E-Commerce"
_ICLR.cc/2026/Conference — Submitted to ICLR 2026_

### Official Review · Reviewer_E77D · 2025-10-29

**Soundness:** 2
**Presentation:** 2
**Contribution:** 2
**Rating:** 2
**Confidence:** 4

**Summary:**

This paper proposes CascadeAgent, a multi-agent framework to automatically generate and optimize LLM prompts for tens of thousands of heterogeneous product attributes on e-commerce platforms. Specifically, the framework includes a scalable "Multi-pass Prompt Generation" (MPG) strategy and a 5-agent collaborative team (CascadeAgent) that iteratively refines the prompts using semantic gradient-based optimization. The paper provide a theoretical proof of the framework's convergence and demonstrate CascadeAgent's effectiveness in large-scale experiments, particularly its ability to significantly boost the performance of low-cost models.

**Strengths:**

1. The paper proposes a practical and highly scalable prompt optimization solution, effectively optimizing over 27,000 heterogeneous attributes.

2. The proposed CascadeAgent method demonstrates significant effectiveness by boosting the more cost-effective Mistral NeMo from 57.14% to 90.21%, closing to the performance of SOTA models Claude 3.5 Sonnet.

3. The paper provides a theoretical analysis that well support the design of the agent framework (although the theoretical analysis is a generally broad analysis that prove the effectiveness of most similar framework).

**Weaknesses:**

1. The experimental evaluation is weak and lacks effective baseline comparisons. The paper only compares CascadeAgent against a "single universal prompt" and "MPG", while no existing, SOTA or previous automatic prompt optimization methods (e.g., APE, APO) are included as baselines, making the experiments less valid.

2. The proposed agent collaboration framework is not novel; similar ideas have been proposed in many prior works [1, 2]. Likewise, the "Optimization with Textual Gradients" section only adopts methods from previous work [3].

3. The proposed multi-agent framework lacks validation of its effectiveness. For instance, the paper claims the Flaw Agent can “move beyond simple error counts, identifying systematic problems, common error patterns” and produce “actionable critiques.” However, there is no ablation study or any case study to support the claim. Similarly, the effectiveness and claimed advantages of the framework's other components are also not proven.


[1] AgentVerse: Facilitating Multi-Agent Collaboration and Exploring Emergent Behaviors.

[2] (Perhaps) Beyond Human Translation: Harnessing Multi-Agent Collaboration for Translating Ultra-Long Literary Texts

[3] Automatic Prompt Optimization with "Gradient Descent" and Beam Search

**Questions:**

See weakness.

---

> ### Author Response · Authors · 2025-11-21
> **Answers to the weaknesses**
>
> - **Evaluation and baseline comparisons**
>
>   **Answer:** We appreciate the concern. CascadeAgent is in the same family as feedback-driven APO methods [3]: we also use textual “gradients” from error cases to refine prompts. Our main difference is Top-K selection: instead of standard beam/greedy selection, we maintain a growing pool of candidate instructions and select Top-K from the entire pool each iteration, with guarantees that the best prompt so far is preserved (Proposition B.1). This is crucial for stability at 27,000+ prompts.
>
>   In early development, we implemented an APO baseline and found that many prompts “overshot”: successive edits often moved too far, causing performance to oscillate or degrade. This motivated our pool-based Top-K scheme and convergence analysis. We acknowledge we should have reported this baseline and will add results in a revised version.
>
>   We also conducted preliminary experiments with APE-style methods. We found (i) a task mismatch: product attribute generation must remain tightly grounded in catalog specifications, but APE-like candidate generation produced prompts that looked good on small hold-out sets yet were unstable across our diverse product distribution; (ii) a scale mismatch: APE is designed for single-prompt optimization, while we must manage 27,000+ prompts, making full APE-style candidate generation and scoring loops prohibitively expensive; and (iii) a methodological mismatch: purely score-based selection sometimes favored candidates that drifted from catalog constraints when evaluated more broadly.
>
>   Our reported baselines are chosen to isolate our core contributions: Universal vs. MPG shows the value of attribute-specific specialization (+21% to +33% precision), and MPG vs. CascadeAgent isolates the benefit of iterative multi-agent refinement (+15% additional improvement on average).
>
> - **Novelty**
>
>   **Answer:** We agree that CascadeAgent builds on prior multi-agent and textual-gradient ideas and clarify our novel aspects:
>
>   1. **MPG:** A framework for prompt engineering at industrial scale (27,000+ product attribute pairs), providing a systematic architecture to manage specialized prompts rather than single-task multi-agent collaboration as in [1, 2].
>   2. **Attribute-specific initialization:** An initial cascade run uses attribute metadata to generate product-specific prompts before refinement. Preliminary experiments starting from generic prompts required many more iterations and often failed to reach comparable performance.
>   3. **Adapted Top-K selection:** Inspired by [3] but modified to maintain a pool over all past candidates. This avoids catastrophic drops (e.g., discarding a 90% accuracy prompt at iteration K and being stuck at 80% afterwards), which we observed with standard beam search in our work.
>   4. **Theoretical guarantees:** Our MDP-based analysis (Theorem 1) and Proposition B.1 provide convergence and stability guarantees tailored to this multi-agent, multi-prompt setting.
>   5. **Industrial-scale validation:** We adapt textual gradient optimization to production scale with 27,000+ prompts and 304,847 labels.
>
>   We acknowledge that our work extends ideas from [1, 2, 3], but the combination of MPG, attribute-specific initialization, adapted Top-K selection, theory, and large-scale deployment constitutes a novel framework for industrial prompt engineering.
>
> - **Validation of the multi-agent framework’s effectiveness.**
>
>   **Answer:** We acknowledge that finer ablations would strengthen the paper and summarize our evidence:
>
>   - **Current ablations:** Appendix A.1 reports ablations on Top-K, error example count, and minibatch size. Table 1 compares CascadeAgent against baseline and vanilla MPG, showing incremental gains attributable to the multi-agent refinement loop.
>   - **Flaw vs. Writing Agent design:** Early in development we tried a one-step design where a single agent saw error cases with ground-truth labels and directly rewrote prompts. This often led to prompts encoding default attribute values from label distributions instead of fixing reasoning issues. Separating into:
>     - a **Flaw Agent** that sees labeled errors and summarizes systematic problems, and
>     - a **Writing Agent** that only sees these summaries (not label distributions) and updates prompts
>     reduced overfitting to specific label distributions and produced more generalizable improvements.
>
>   - **Empirical evidence:** At scale, CascadeAgent yields consistent gains over 27,000+ attribute pairs, with +21% to +33% precision improvements across models and 58.2% of high-impact PAs improving on both precision and coverage. Appendix A3 provides concrete case studies where Flaw Agent summaries (e.g., “inconsistent style categorization”) lead to actionable prompt refinements.
>
>   We agree that additional ablations isolating each agent (e.g., removing the Flaw Agent while keeping others) would provide further insight and consider this important follow-up work.

---

### Official Review · Reviewer_FP4w · 2025-11-01

**Soundness:** 2
**Presentation:** 2
**Contribution:** 2
**Rating:** 2
**Confidence:** 4

**Summary:**

This is more like a technical report than a research paper.

**Strengths:**

The prompt rewriting task is very interesting.

**Weaknesses:**

I read the paper carefully, I dont think the paper meets the high standard for publishing in ICLR. The theory part is too simple, not too much contribution, the result section is too short, and overall paper lacks clear contribution. Not ICLR level.

**Questions:**

Would suggest authors to address issues in weakness.

---

### Official Review · Reviewer_dZh1 · 2025-11-01

**Soundness:** 3
**Presentation:** 3
**Contribution:** 3
**Rating:** 4
**Confidence:** 4

**Summary:**

This paper addresses the formidable challenge of automating prompt engineering for domain-specific tasks at an industrial scale, with a focus on e-commerce product attribute generation involving tens of thousands of distinct attributes. The authors propose CascadeAgent, a novel multi-agent framework designed to automate the creation and iterative refinement of attribute-specific prompts. The system is built upon two key ideas:  Multi-pass Prompt Generation (MPG), which modularizes the problem by decomposing catalog enrichment into attribute-specific sub-tasks, and a collaborative loop of five specialized agents (Prompting, Writing, Generation, Evaluation, Flaw Detection) that refine prompts using a semantic gradient-based optimization strategy. A significant contribution of this work is the provision of a formal theoretical analysis, modeling the refinement process as a Markov Decision Process and proving convergence towards reduced catalog loss under defined assumptions.

**Strengths:**

1.This paper exhibits strengths particularly in its theoretical grounding. The core originality lies in the creative integration of a multi-agent architecture with a formal convergence guarantee to address the underexplored challenge of large-scale prompt optimization. This represents a significant conceptual leap beyond prior work that typically handles dozens or hundreds of prompts. The theoretical contribution is substantial; the authors provide a rigorous Markov Decision Process model and a formal proof of convergence, which elevates the work from a purely engineering solution to a principled method with provable properties.

2.The significance of the work is underscored by its successful application in a real-world industrial setting, managing an unprecedented scale of over 27,000 distinct prompts. The empirical results are compelling, demonstrating that the framework not only improves performance metrics significantly but also makes cost-effective models viable for deployment, a finding with direct practical implications.

**Weaknesses:**

1.The description of the core multi-agent workflow remains high-level and ambiguous. Figure 1 is too simplistic and does not elucidate the exact information flow, data structures, or prompting protocols that govern the interactions between the five specialized agents. For instance, the specific input and output formats for the Writing Agent and how it synthesizes the Flaw Agent's feedback into a revised prompt are left entirely to the reader's imagination.

2.Despite the impressive scale of the experiments, the evidence presented is not fully convincing. The performance tables report aggregate scores but lack illustrative case studies. There is no qualitative analysis showing how a problematic prompt for a specific attribute was iteratively refined by the cascade into a high-performing one, which would have been crucial for demonstrating the system's operational effectiveness and value beyond mere metrics.

3.The authors explicitly state that due to company policy, they are unable to release the code and data, which limits readers' ability to conveniently verify and reproduce the results.

**Questions:**

1.Could you provide a more detailed flowchart specifying the exact inputs, outputs, and data formats for each of the five agents in a single iteration?

2.What is a concrete example of an initial prompt, the corresponding flaw summary from the Flaw Agent, and the resulting optimized prompt for a specific product attribute?

3.In the theoretical analysis, how was the key assumption of the "marginal-churn condition" (p_fix > p_break) validated or estimated in your practical setting?

4.Given the inability to release code, could you provide a pseudo-code snippet illustrating the core orchestration logic of the multi-agent loop?

---

> ### Author Response · Authors · 2025-11-20
> **Answers to each question**
>
> - **Could you provide a more detailed flowchart specifying the exact inputs, outputs, and data formats for each of the five agents in a single iteration?**
>
>   **Answer:** Thank you for this suggestion. We have added Figure 8 in Appendix A2, which illustrates the iterative prompt refinement process and shows how the five agents interact. Additionally, Appendix A2 provides detailed descriptions of the prompt structure and data formats used throughout the system, including:
>
>   - System role, product context, task definition, attribute-specific instructions, and output schema
>   - Figure 7 showing the prompt generation process for the initial cascade run by the Writing Agent
>   - The specific data exchanged between agents during refinement
>
>   While Figure 1 in the main text provides the high-level architecture, Figure 8 and the accompanying descriptions in Appendix A2 detail the iterative workflow.
>
>
>
> - **What is a concrete example of an initial prompt, the corresponding flaw summary from the Flaw Agent, and the resulting optimized prompt for a specific product attribute?**
>
>   **Answer:** Thank you for this suggestion. We have added Appendix A3, which provides three representative examples (BACKPACK – strap_type, FACIAL_TISSUE – item_form, and CANDLE – style) showing the transformation from initial instructions to optimized prompts, along with corresponding performance improvements.
>
>
>
> - **In the theoretical analysis, how was the key assumption of the "marginal-churn condition" (p_fix > p_break) validated or estimated in your practical setting?**
>
>   **Answer:** Thank you for this question about our theoretical assumptions. Our results strongly suggest that the marginal-churn condition holds in practice:
>   - 96.6% of PAs avoided dual-metric degradation (consistent with p_fix > p_break)
>   - 58.2% showed win–win improvements (strong net benefit)
>   - Only 2.4% were complete failures (rare violation of the assumption)
>
>   The marginal-churn condition (p_fix > p_break) was a foundational assumption underlying our approach and reflects common sense about our system design: when a Flaw Agent explicitly identifies errors (e.g., “incorrectly predicting ‘plastic’ for products without material information”) and a Writing Agent refines the instruction to address this specific issue, it would be surprising if such targeted corrections systematically made things worse. The assumption captures the intuition that deliberate, error-informed refinements are net beneficial. Although we did not directly estimate p_fix and p_break, the practicality of CascadeAgent’s improvements fundamentally depended on this condition holding across diverse attributes, and the empirical outcomes above validate that it did.
>
>
>
> - **Given the inability to release code, could you provide a pseudo-code snippet illustrating the core orchestration logic of the multi-agent loop?**
>
>   **Answer:** Please see the newly added Appendix A2.

---

### Official Review · Reviewer_oNDQ · 2025-11-01

**Soundness:** 3
**Presentation:** 4
**Contribution:** 3
**Rating:** 6
**Confidence:** 4

**Summary:**

The **CascadeAgent** framework introduces a scalable, multi-agent prompt refinement approach (MPG) for large e-commerce attribute generation, combining modular decomposition, iterative semantic optimization, formal convergence analysis, and cross-model efficiency; key strengths (scale, precision/coverage gains, economic viability) coexist with resource intensity and difficulty on complex attributes.

• **Core Contribution**: CascadeAgent—industrial-scale multi-agent framework for prompt adaptation and specialization.

• **Scalability Mechanism**: Multi-pass Prompt Generation (MPG) decomposes >27,000 attribute-specific sub-tasks for independent optimization.

• **Architecture**: Five agents (Prompting, Writing, Generation, Evaluation, Flaw Detection) iteratively refine via semantic gradient-based optimization.

• **Theoretical Rigor**: Formal analysis with convergence toward reduced catalog loss $L(\pi)$ (Markov Decision Process framing).

• **Empirical Performance**: Up to +33% precision and +14% coverage improvements across multiple LLMs.

• **Economic Impact**: Elevated Mistral NeMo—precision gap narrowed to ~3% vs premium Claude 3.5 Sonnet.

• **Complex Attribute Limits**: Challenges remain for image-dependent and numeric attributes despite general scalability.

• **Resource Requirements**: High compute (48 H100 GPUs) and substantial labeled ground truth needed for optimization.

**Strengths:**

The paper presents a novel and significant contribution to prompt engineering by introducing **CascadeAgent**. The originality stems from the principled integration of multi-agent systems with a scalable decomposition strategy called **Multi-pass Prompt Generation (MPG)**, enabling optimization of large number of attribute-specific prompts.

• **Novel Framework** and Scale: CascadeAgent’s architecture manages and optimizes over 27,000 attribute-specific prompts, surpassing prior approaches limited to dozens or hundreds. The framework is designed for industrial-scale product attribute generation in e-commerce.

• **Theoretical Guarantees**: Formal analysis models prompt refinement as a Markov Decision Process (MDP) and demonstrates convergence towards reduced catalog loss. Empirical results show +33% precision and +14% coverage gains while enabling a cost-effective model (Mistral NeMo) to approach premium performance.

• **Modularity** and Specialization: MPG enables modular decomposition and independent attribute optimization; five specialized agents (Prompting, Writing, Generation, Evaluation, Flaw Detection) drive nuanced improvements..

• Effective **Optimization**: Semantic gradient-based (textual gradients) iterative refinement achieved an average +15% improvement in hold-out test accuracy for Product-Attributes needing tuning.

• **Economic Viability**: The framework elevated Mistral NeMo, reducing the precision performance gap with Claude 3.5 Sonnet to only 3%.

• Seamless **Extension**: Multi-agent architecture allows adding new attributes without complete retraining, addressing prior limitations.

• Robust **Validation**: Industrial-scale evaluation using a 10,000-product catalog set and a high-fidelity set of 1,879 Product-Attribute pairs.

• Improved **Interpretability**: Iterative agent cascade steps provide insight into how and why prompts are modified.

**Weaknesses:**

While CascadeAgent demonstrates **significant advancement**, several limitations related to its scope, computational requirements, and remaining performance challenges need constructive consideration.

• **High Computational Requirements**: The multi-agent system demands substantial computing infrastructure, utilizing 48 NVIDIA H100 GPUs across 6 AWS EC2 P5 instances for parallel processing, making it potentially inaccessible in resource-constrained environments.

• **Limited Domain Validation**: The empirical validation is confined primarily on e-commerce attribute enrichment, and the transferability to other domains remains untested.

• **Dependence on Labeled Data**: The iterative optimization process relies heavily on ground truth data availability, requiring a minimum of 150 verified human labels per PA for the high-fidelity optimization set. Future work should explore semi-supervised or agent-driven data synthesis to mitigate this reliance which otherwise could pose a challenge in data-scarce domains.

• Challenges with **Complex Attribute** Types: The optimization process and system performance plateaued for a significant percentage of PAs, particularly those that were image-dependent attributes (40% of remaining cases) and numeric attributes, indicating a need for multimodal models or enhanced mathematical reasoning capabilities to mitigate these limitations in the current LLM instruction refinement loop.

• Trade-off Issues (**Precision vs. Coverage**): A large portion of PAs (35.0%) exhibited increased coverage but reduced precision post-refinement, often due to hallucinations. The refinement strategy may need adjustments to better control precision loss in exchange for coverage gains.

• **Hinderance to Reproducibility**: Although some details/specifications are provided, the implementation code and dataset cannot be released due to company policy, which presents an obstacle to complete independent verification and replication. Moreover, there aren't any experiments conducted on similar public datasets.

**Questions:**

The **CascadeAgent framework** raises interrelated concerns across dataset transparency, **generalizability**, model selection scope, reproducibility of sampling and optimization settings, cost vs. claimed affordability, observed **optimization plateau** for complex attributes, sensitivity of convergence to loss weighting, and empirical validation of theoretical sample size assumptions.

1. **Dataset transparency**: Beyond counts (10,000 products; 1,879 high-fidelity PAs), can synthetic examples (titles/descriptions + initial and refined prompts) be shared to illustrate **prompt structure** and attribute complexity?

2. **Domain generalizability**: Why no evaluation on public e-commerce attribute datasets (e.g., OpenTag, AdaTag, MAVE) to test **transferability** beyond the internal case study?

3. **Model selection scope**: Why limit evaluation to Mistral NeMo (generation) and Claude 3.5 Sonnet (flaw detection); why exclude other open-source or proprietary models that could strengthen affordability claims?

4. **Sampling reproducibility**: Hyperparameters (Top-K, minibatch) are given, but missing temperature / nucleus sampling for Generation & Writing Agents—can these stochastic **sampling parameters** be disclosed?

5. **Cost-benefit justification**: With 48 H100 GPUs (6 P5 instances), can a comparative **cost analysis** show the cascade’s total inference + premium calls outperforming alternatives (supervised fine-tuning or single premium zero-shot)?

6. **Complex attribute plateau**: For unmet stopping criteria (46.0% PAs; 40% image-dependent), what concrete plans exist to integrate multimodal or enhanced **mathematical reasoning** into the semantic gradient loop?

7. **Convergence sensitivity**: How do precision/coverage outcomes and convergence rates vary under adjustments to loss weights ($w_{val}, w_{omis}, w_{commis}$) influencing $r = \kappa_{\max}/\kappa_{\min}$ and contraction factor $\phi$; were weight sweeps tested to reduce the 35.0% precision drop cases?

8. **Sample size validation**: For Proposition 1, how was evaluation sample size $n$ tied to the dynamic loss gap $\Delta_t$ computed or verified across 1,879 PAs—was $n$ adaptively scaled or empirically stress-tested to support monotone improvement guarantees?

**Details Of Ethics Concerns:**

- Although the **ethics statement** is clear, this study relied on human evaluations of product attributes.
- Based on comments from other reviewers and if deemed necessary by Area Chairs, an additional review would be meaningful

---

> ### Author Response · Authors · 2025-11-20
> **Answers to the 8 questions**
>
> We thank the reviewer for the suggestions.
>
> - **Dataset transparency:** Yes. Appendix A2 gives the prompt architecture (system role, product context, task, attribute instructions, output schema) and the generation/refinement flow (Figures 7–8). Appendix A3 shows examples with initial vs. optimized prompts and precision/recall gains on test data (≥100 labels).
>
> - **Domain generalizability:** OpenTag, AdaTag, and MAVE are supervised extraction benchmarks for values present in text, whereas CascadeAgent automates prompt engineering for attribute generation, including missing/implicit values. Our method needs attribute metadata (definitions) that these datasets lack and they evaluate extraction accuracy rather than prompt optimization. We acknowledge that creating public benchmarks for automated prompt engineering would be valuable future work; here we focus on production deployment.
>
> - **Model selection scope:** Our model choices follow production constraints. Generation and evaluation use Mistral NeMo for its cost-latency-quality trade-off and licensing for 27,000+ product attribute pairs. The Writing and Flaw Detection agents use Claude 3.5 Sonnet V2, which performed best in comparisons for iterative critique and refinement. Our key claim is that CascadeAgent’s prompt optimization enables a smaller, more affordable model to approach a premium model's precision and should extend to other model pairs.
>
> - **Sampling reproducibility:** For Generation and Evaluation (Mistral NeMo) we use temperature = 0 and top-p = 0.9 (default), giving deterministic outputs. For Writing and Flaw Detection (Claude 3.5 Sonnet V2) we use temperature = 0.3 and top-p = 0.999 (default), allowing mild wording variation. Randomness is mainly in error-case sampling, not prompt rewrites, so the semantic gradient loop learns from diverse failures while producing stable prompt improvements.
>
> - **Cost-benefit justification:** CascadeAgent anchors a tiered pipeline: (1) with optimized prompts, Mistral NeMo handles most traffic and reaches within 3% of Claude 3.5 Sonnet’s precision; (2) product-attributes that still miss business thresholds are escalated to Claude 3.5 Sonnet using the same optimized prompts; (3) premium outputs provide labels to fine-tune smaller models on hard cases, reducing premium dependence. Compared with fine-tuning alone, CascadeAgent’s optimized prompts improve all models they are paired with and provide richer context for targeted fine-tuning. The 48 H100 GPUs support serving, fine-tuning, and prompt optimization; prompt optimization is a one-time cost whose benefits persist across tiers.
>
> - **Complex attribute plateau:** For PAs that do not meet the stopping criteria, image-dependent attributes are selectively routed to multimodal LLMs while others remain on text-only models such as Mistral NeMo, and multimodal Claude models are used in the Flaw Detection and Writing agents so the semantic gradient loop reasons over text and images. Numerically complex attributes are routed to models with stronger quantitative reasoning and fine-tuned on mathematical reasoning tasks using premium outputs. When performance plateaus after ~3 iterations (indicating a model capability ceiling), we stop further prompt iterations and instead introduce multimodal inputs (when relevant), escalate to larger models, or initiate targeted fine-tuning.
>
> - **Convergence sensitivity:** The 35% figure denotes PAs where CascadeAgent increased coverage while reducing precision relative to baseline (Quadrant III in Figure 2), often for sparse attributes such as “subject_character,” where filling more blanks increases hallucinations. We used uniform weights (w_val ≈ w_omis ≈ w_commis ≈ 0.33) to stay neutral between coverage- and precision-oriented stakeholders, to obtain the fastest contraction rate with r = κ_max/κ_min = 1.0 in Theorem 1, and to keep configuration simple. We did not sweep weight configurations; attribute-specific weights (e.g., higher w_commis for sparse attributes) could reduce Quadrant III but would slow convergence (higher r → worse φ) and complicate stakeholder alignment. Uniform weights are globally optimal for contraction but locally suboptimal on heterogeneous attributes.
>
> - **Sample size validation:** We used a fixed validation budget of n = 100 samples per PA across 1,879 attributes and did not adapt n to the observed Δt. Proposition 1’s requirement n = Θ(Δt^{-2} log(|P_t|/ϑ)) serves as design guidance rather than a strict constraint; with n = 100, the guarantees are most relevant when Δt ≳ 0.15–0.20. We design the cascade to generate meaningfully different candidate prompts so that loss gaps are large enough for reliable Top-K selection under this fixed budget. Empirically, 58.2% of PAs achieved win–win improvements, 96.6% avoided dual-metric degradation, and only 2.4% showed clear failure patterns, indicating catastrophic selection errors from insufficient n were rare in practice.

---

### Meta-Review · Area_Chair_RYrj · 2026-01-07

**Summary:**

This paper presents CascadeAgent, a multi-agent framework for large-scale automated prompt engineering, motivated by industrial e-commerce attribute generation. The work addresses an important and practical problem. Reviewers generally agree that the problem setting is relevant and that the system shows promise in a real-world context. However, the reviewers also identify several significant issues that limit the strength of the submission.

A primary concern is the clarity and strength of the contribution, particularly with respect to novelty and theoretical depth. Multiple reviewers note that the proposed framework builds heavily on prior work in multi-agent collaboration and textual-gradient-based prompt optimization, with limited differentiation beyond engineering scale and system integration.

Reviewers also raise concerns about the experimental evaluation and validation. The experiments are confined to a single internal e-commerce dataset, with no evaluation on public benchmarks (especially given neither code nor data can be released due to company policy). This makes it difficult to assess generalizability. Several reviewers highlight the lack of strong baseline comparisons, particularly against existing automatic prompt optimization methods.

Overall, while the paper demonstrates a strong applied system and valuable engineering experience at scale, the reviewers agree that clearer articulation of novelty, stronger empirical validation against established baselines, and broader evaluation, would be required to support acceptance at ICLR.

**Reviewer Concerns:**

I believe that most reviewers’ core concerns remain outstanding. Specifically,

Reviewer oNDQ's concerns about domain generalization, dependence on labeled data, precision–coverage trade-offs, computational cost, and reproducibility, would likely remain.

Reviewer dZh1's concerns about ambiguity in the multi-agent workflow, lack of qualitative examples, and limited reproducibility, would likely remain.

Reviewer FP4w's concerns on weak theory and unclear contribution will remain as no engagement in the rebuttal phase.

Reviewer E77D's concerns about novelty, missing baselines, lack of ablations validating agent roles, and incremental reuse of prior methods, are not fully resolved.

**Reviewer Scores:**

Across reviewers, a full discussion would likely result in at most small positive score adjustments, primarily driven by improved clarity, additional explanations, and acknowledgment of limitations in the rebuttal. However, many of the key concerns, particularly regarding novelty, experimental rigor, generalizability, and reproducibility, are not fully addressed and would require substantive revisions beyond rebuttal alone.

---

### Decision · Program_Chairs · 2026-01-26

Reject